Registered report: stage 2

Subject Areas:
ecology/environmental science

Keywords:
conservation, endangered species, policy signal, survival analysis, large carnivore, *Canis lupus baileyi*

Author for correspondence:
Naomi X. Louchouarn
e-mail: louchouarn@wisc.edu

†Equal first co-authors.

# Evaluating how lethal management affects poaching of Mexican wolves

Naomi X. Louchouarn[1,†], Francisco J. Santiago-Ávila[1,2,†], David R. Parsons[2] and Adrian Treves[1,2]

[1]Nelson Institute for Environmental Studies, University of Wisconsin-Madison, Madison, WI, USA
[2]Project Coyote Science Advisory Board, Larkspur, CA, USA

 NXL, 0000-0003-0047-8808; FJS-Á, 0000-0003-4233-9128

Despite illegal killing (poaching) being the major cause of death among large carnivores globally, little is known about the effect of implementing lethal management policies on poaching. Two opposing hypotheses have been proposed in the literature: implementing lethal management may decrease poaching incidence (killing for tolerance) or increase it (facilitated illegal killing). Here, we report a test of the two opposed hypotheses that poaching (reported and unreported) of Mexican grey wolves (*Canis lupus baileyi*) in Arizona and New Mexico, USA, responded to changes in policy that reduced protections to allow more wolf-killing. We employ advanced biostatistical survival and competing risk methods to data on individual resightings, mortality and disappearances of collared Mexican wolves, supplemented with Bayes factors to assess the strength of evidence. We find inconclusive evidence for any decreases in reported poaching. We also find strong evidence that Mexican wolves were 121% more likely to disappear during periods of reduced protections than during periods of stricter protections, with only slight changes in legal removals by the agency. Therefore, we find strong support for the 'facilitated illegal killing' hypothesis and none for the 'killing for tolerance' hypothesis. We provide recommendations for improving the effectiveness of US policy on environmental crimes, endangered species and protections for wild animals. Our results have implications beyond the USA or wolves because the results suggest transformations of decades-old management interventions against human-caused mortality among wild animals subject to high rates of poaching.

# 1. Background

Human-caused mortality is the major cause of death among large, terrestrial, mammalian carnivores worldwide [1], including the

USA [2–5]. Anthropogenic mortality has precipitated the decline and extirpation of carnivore populations worldwide both indirectly and directly through the often coinciding threats of habitat loss and degradation, prey depletion and killing [6]. Indeed, reported and unreported poaching is the major form of human-caused mortality for large carnivore populations in several regions [7,8], including five US wolf populations [4]. Such mortality raises individual and societal concerns because poaching is an environmental crime, harms individual animals, and undermines restoration and conservation efforts.

Identifying and estimating poaching is hindered by concealment of evidence. Estimating concealed, illicit killing rates has recently been transformed by two analyses that used data that had previously been ignored. Liberg et al. [7] estimated the hazard rate of cryptic (i.e. unreported or concealed) poaching by considering slow-downs in population growth and accounting for the disappearances of marked grey wolves in Scandinavia. Similarly, Treves et al. [5] re-calculated the risk of poaching relative to other causes of death by considering missing, marked animals, which had previously been excluded from analyses under an erroneous assumption that marked animals that disappeared would have died of similar causes as those marked animals found dead. Therefore, investigators can now better estimate heretofore under-appreciated variables that are essential to understanding population dynamics and individual animal life histories. However, the latter study admittedly did not directly estimate poaching, instead using estimates from other populations (Scandinavia and Wisconsin) as multipliers to indirectly quantify cryptic poaching, and did not measure policy effects on poaching or consider time to exposure of wolves to policies. Its objective was strictly to estimate the risk of poaching in a population regardless of policy period. Therefore, here, we propose an important advance to estimate the relationships between policy interventions and fates of marked carnivores, while controlling for spatio-temporal covariates. We test opposed hypotheses from the literature explained next.

The scientific literature has recently addressed the question of if and how policies may influence the hazard and incidence of poaching. The usual assumption (despite lack of empirical evidence) is that some predator-killing (e.g. government permits for killing or public hunting seasons) might increase tolerance for a species (and thus reduce poaching); an argument first articulated in federal court in 2006 [9] and summarized more generally in [10]. We call this first hypothesis 'killing for tolerance', which predicts legal killing will reduce poaching through the following mechanism: legalizing or liberalizing killing of controversial species will lead would-be perpetrators to desist from poaching because of increased tolerance for the species or approval for protectionist policies. Early tests of this notion of 'killing for tolerance' include [11–17]. Olson et al. [11] examined correlations between documented (i.e. reported) poaching of Wisconsin's wolves and management policies between 2003 and 2011. They suggested that the incidence of known poaching events was inversely related to the proportion of each year with state management associated with liberalized killing periods, and hypothesized that frustration with protections for wolves led to increased poaching. Studying the same population, albeit with more sophisticated modelling of demographic processes, Stenglein et al. [18] estimated an additional mortality of 4% was necessary to explain the observed slow-down in the population's annual growth rate within that same time period. These early tests of the killing for tolerance hypothesis attribute rates of poaching and population dynamic changes to illegal actions motivated by inconsistent management and protections for controversial wolves. By contrast, Chapron & Treves [15] reported serial slow-downs of wolf population growth during six non-consecutive policy periods in Wisconsin and Michigan from 1995 to 2012, which seemed attributable to unreported wolf-killing. They proposed an explanation we refer to as 'facilitated illegal killing'. Three social scientific studies published between 2013 and 2015 [12–14] examined attitudes towards wolves in Wisconsin and found that tolerance decreased as wolf-killing was progressively liberalized, or intention to poach wolves increased as wolf-killing was progressively liberalized from 2003 to 2013. Considering such evidence, the alternative hypothesis of 'facilitated illegal killing' suggests that liberalized killing might decrease the value of wolves to would-be perpetrators of poaching, or decreasing the risk of being caught [15]. A 2019 re-analysis using the methods proposed below found liberalized killing policy periods in Wisconsin, USA (1979–2012), were associated with increases in hazard and incidence of wolf disappearances that outweighed by fivefold any decreases in reported poaching, undercutting the 'killing for tolerance' hypothesis [19]. Despite the lack of a clear causal connection between attitudes and poaching, the study described here tries to establish a closer mechanistic link between policies and poaching behaviour. In sum, two published hypotheses make opposed predictions about the rates of poaching in relation to policies for liberalizing legal killing of controversial species.

Other research linking wolf mortality to population growth rates in a hunted Finnish population found increases in population size were positively associated with increases in poaching [8]. Using generalized linear models focused on predictors of poaching, the same team later found the number

of legally hunted wolves both across the country and at the local scale was associated with a decrease in the probability of poaching, while increases in the number of wolves that could be legally killed (the 'bag limit') were associated with increases in the probability of poaching [20]. Additionally, the authors suggest that declines in poaching following higher levels of legal hunting might be an artefact of a decrease in the individual wolves exposed to poaching [20, p. 7]. They concluded that 'tolerance for carnivores cannot be promoted by legal hunting alone, so more comprehensive conservation efforts are needed' [8,20].

The most recent publication on this topic for grey wolves in Scandinavia [21] suggested that when more territorial breeding individuals were removed legally, fewer such animals disappeared (presumably poached), but their analysis has been questioned on the grounds of inappropriate statistical analyses and incomplete treatment of the apparent rise in disappearances during years with legal wolf-killing [22].

However, there remain unresolved concerns about omitted methods and the statistical approaches and assumptions made in all three studies from the US upper Midwest and Nordic countries [19,23]. None of the studies [8,15,16,18] explicitly modelled survival in relation to the amount of time wolves were exposed to liberalized killing policies that changed 12 times between 1995 and 2012 in the USA and several times in the Nordic countries [15,24]. Here, we build on these analyses by including the amount of time that radio-collared wolves were exposed to liberalized killing policies to re-estimate hazard and incidence of poaching.

Indeed, a simple reduction in poaching may not be equivalent to 'tolerance'; that is, greater acceptance of wolves on the landscape. The 'killing for tolerance' hypothesis suggests a cognitive mechanism that implies something broader than a reduction in poaching, which is only one anthropogenic endpoint affected by tolerance. However, poaching is not the only human behaviour affected by tolerance causing wolf mortality. Human tolerance would arguably affect other anthropogenic endpoints such as legal killings or (in this particular population) final removals (e.g. through increased legal killings by citizens or requests for action to government agents leading to increased mortality, similar to that found in Scandinavia [20]). Thus, any comprehensive exploration of 'tolerance' affecting wolf mortality should examine the interactions between the different anthropogenic endpoints and their resulting incidences.

Opposing views of the relationship between legal killing and poaching of wolves can be tested if we analyse individual wolf survival in relation to the timing and duration of their exposure to periods with different policies for legal killing. Here, we will test the specific hypothesis that rates of hazard and incidence of mortality or disappearance of wild Mexican grey wolves (Canis lupus baileyi) in Arizona and New Mexico, USA, changed after policies altered the legality of killing or harassment of Mexican wolves by the public and government agencies. Table 1 provides predictions for the two response variables of hazard ratios (HRs) and competing risk subhazard ratios (SHRs). There has been no research on the opposing hypotheses of 'killing for tolerance' or 'facilitated illegal killing', within this population. Such a test is particularly important as Mexican wolves are a highly endangered subspecies of grey wolves. Previous estimates [4] indicate poaching rates in this population have been high and underestimated by traditional methods.

The subspecies was functionally extinct in the USA by the 1970s due to extermination efforts by state, federal, tribal and private actors [25]. A captive breeding programme began in 1977, and the US government began releasing Mexican wolves to the Blue Range Wolf Recovery Area in southern Arizona and New Mexico in 1998. From 1998 to 2016, all Mexican wolves released to the recovery area were fitted with a radio collar and were closely monitored ($n = 279$ radio-collared). Here, we examine the survival and disappearances of adult marked Mexican wolves before, during and after two policy periods, one in 2005–2009 and another starting in 2015, that liberalized killing or removal of wolves by government or private actors.

We examine data on radio-collared animals using competing risk analyses that allow the modelling of hazards and incidences of multiple fates (i.e. various causes of death or lost to monitoring; *endpoints*, hereafter) while controlling for multiple covariates. We model exposure time to policy changes. This analysis allows us to make inferences beyond the cursory examination of compensatory mortality causes due to changes in policy, to examine how policies might affect an individual wolf's probability of succumbing to a cause of death. Furthermore, using a competing risks analysis allows us to include disappearance as an endpoint which is crucial, given that prior work shows that censoring lost to follow-up (LTF) led to systematic underestimates of poaching in other grey wolf populations [4,19]. The results of this analysis can provide recommendations for improving the effectiveness of US policy on environmental crimes, endangered species and protections for wild animals. Therefore, our analyses have implications beyond the USA or wolves because the methods promise to transform

**Table 1.** Relationship between our hypotheses, proposed analyses and interpretation of outcomes (including contingent interpretation and synthesis of model results). $HR_{poa}$ refers to the HR of the poaching endpoint, while $HR_{ltf}$ refers to the HR of the lost to follow-up endpoint.

| question | hypotheses | sampling plan (e.g. power analysis) | analysis plan | interpretation given different outcomes (see note and main text for Bayes factor specifications[a]) |
|---|---|---|---|---|
| Do hazard rates or cumulative incidence of death by poaching or disappearance (DV) of wild, collared adult Mexican grey wolves change after policies change (IV) from strict protection to liberalized killing and back again? | 'Killing for tolerance' predicts the hazard and incidence decline for the endpoint 'poached' (poa) or the endpoint LTF when the IV of policy period liberalizes wolf-killing. | All collared wild Mexican grey wolves from MWRP and OLE 1998–2016 ($n = 279$). A diagnostic test is run on the samples with four endpoints (human, natural, removal, LTF) before proceeding to the analysis plan (see Diagnostic step). See tables 3 and 4 for endpoint-specific sample sizes split by the IV of policy period. | For MWRP and OLE datasets: Endpoint-specific Cox multiple regression models (for each endpoint) on the IV of policy period and other covariates. Competing risk Fine and Gray multiple regression models (for each endpoint) on the IV of policy period and other covariates. CIFs allow for analysis of population effects (incidence) while considering the prevalence of each endpoint in the population. | $HR_{poa}$ and $HR_{ltf}$ are <1 or ($HR_{poa}$ has to be <1 and greater in magnitude than any increase in $HR_{ltf}$ or $HR_{ltf}$ has to be <1 and greater in magnitude than any increase in $HR_{poa}$ and endpoint-specific CIFs estimate which endpoint has a greater effect on the population (from Fine–Gray models of competing risks). The criterion for determining if 'TOTAL potential poached' probability $f$ declined is a decline in the combined incidence of LTF and POA. |

**Table 1.** (Continued.)

| question | hypotheses | sampling plan (e.g. power analysis) | analysis plan | interpretation given different outcomes (see note and main text for Bayes factor specifications[a]) |
|---|---|---|---|---|
| | 'Facilitated illegal killing' predicts the hazard and incidence increase for the endpoint 'poached' (poa) or the endpoint LTF when the IV of policy period liberalizes wolf-killing. | All collared wild Mexican gray wolves from MWRP and OLE 1998–2016 ($n = 279$). A diagnostic test is run on the samples with four endpoints (human, natural, removal, LTF) first before proceeding to the analysis plan (see Diagnostic step). See tables 3 and 4 for endpoint-specific sample sizes split by the IV of policy period. | For MWRP and OLE datasets: Endpoint-specific Cox multiple regression models (for each endpoint) on the IV of policy period and other covariates. Competing risk Fine and Gray multiple regression models (for each endpoint) on the IV of policy period and other covariates. CIFs allow for analysis of population effects (incidence) while considering the prevalence of each endpoint in the population. | $HR_{poa}$ and $HR_{ltf}$ are >1 OR ($HR_{poa}$ has to be >1 and greater than any decrease in $HR_{ltf}$ OR $HR_{ltf}$ has to be >1 and greater than any decrease in $HR_{poa}$) AND endpoint-specific CIFs estimate which endpoint has a greater effect on the population (from Fine–Gray models of competing risks). The criterion for determining if 'TOTAL potential poached' probability for wolves declined is a decline in the combined incidence of LTF and POA. |

[a]Following reviewer recommendations, we will use BF using three specifications. BF estimates the strength for our alternative and null hypotheses for particular endpoints, and allows us to assess insensitivity of the data to resolve differences between hypotheses. For purposes of comparison, and to provide estimates of policy effects on 'total potential (cryptic+reported) poaching', we proceed to aggregate poaching endpoints and run all analysis on the new endpoint LTF + POA (including BFs) (see *Statistical methods*).

scientific understanding of processes and patterns in human-caused mortality among wild animals subject to high rates of unregulated killing.

# 2. Methods

## 2.1. Data collection and preparation

We analysed data acquired from the Department of Interior US Fish and Wildlife service (USFWS) Mexican Wolf Recovery Program (MWRP) and their Office of Law Enforcement (OLE) in separate but overlapping datasets on marked (hereafter collared), monitored Mexican wolves in the wild. The MWRP survival data include the monitoring history for all collared and monitored adult Mexican wolves in the wild since the beginning of the recovery programme, 29 March 1998–31 December 2016; $n = 279$ (monitored wolf pups were excluded from this dataset).

Because of the small wild population size and the captive breeding programme, the majority of wolves in the Mexican wolf recovery area were collared for monitoring. Only wild-born individuals that eluded capture remained unmonitored. Therefore, our analysis has the benefit of reducing (but not completely eliminating) a common bias in monitored animal studies when the marked subsamples are non-random, unrepresentative samples of the wild population and may not have the same mortality risk as unmarked individuals [26–30].

The MWRP survival data contain the following individual covariates we used in our statistical analysis: monitoring start date, sex and endpoint (i.e. end of monitoring time by: cause of death, lost to follow-up (LTF) or removal by agency action). The endpoint 'removal' by agency action typically involved USFWS live-capture of a wolf from the wild followed by either placement in captivity or killing. The endpoint of LTF occurred when the telemetry equipment affixed to a wolf in the wild stopped functioning and the collar was never recovered. This could happen from mechanical/battery failure or destruction by external causes such as humans. Some wolves had multiple collars during their monitoring history as a result of recapture and recollaring. The vast majority of monitored time intervals (87.6%) were obtained using VHF collars, while the remaining 13% of monitored intervals were obtained with GPS collars. In our data, the average amount of time to LTF for wolves wearing VHF collars was 621 days, with a range of 7–3079 days. The average battery life of a VHF radio collar is about 1095 days, but wolves were often recollared. Only one individual disappeared while wearing a GPS collar, and this individual went LTF after 169 days. For recovered collars, the cause of death was estimated by USFWS using standard methods following necropsy and radiography.

For each of the 279 wolves in the MWRP survival data (1998–2016), we estimated the time between collaring (monitoring start date) and endpoint in days ($t$), but we did so differently for surviving, dead and LTF endpoints. We censored surviving, monitored wolves at the end of our study period. For LTF endpoints, we used the date of last telemetry contact. The inclusion of LTF as a separate, explicitly modelled endpoint was crucial for our inferences because of the prior work showing that censoring LTF led to systematic underestimates of the proportion and hazard of poaching in other grey wolves [4,19]. Some wolves might have lived on for a time after their telemetry contact was lost, so LTF represents a systematic underestimate of survival and hence of our parameter, $t$. We address the consequences and magnitudes of that bias in Results. For our 'mortality' and 'agency removal' endpoints, we estimated $t$ for wolves monitored by telemetry until death and the date of final removal to captivity by agency action, respectively.

Mortality endpoints obtained from the MWRP survival data were classified only as 'human' or 'natural' in our first analysis step. Natural used by both MWRP and OLE presumably meant non-human cause of death. The human-caused endpoint was identified in the MWRP data as mortality with 'likely and known human causes', without a more specific cause of death (e.g. vehicle collision, poaching). In the second analysis step, we turned to the OLE data, which categorized human-caused mortality by the following causes of death: vehicle collisions, trap, gunshot, blunt force trauma, 'unknown' or 'other'. Using these data, we classified human-caused deaths as either poached (trap, gunshot, blunt force trauma) or non-criminal (vehicle collisions, 'unknown', 'other' with no evidence of human intent). We used all human-caused deaths recorded up to and including 31 December 2016.

We focused our analysis on a time-varying covariate for policy period (the policy intervention or IV in table 1). Policy period was binary for period of liberalized wolf-killing (1) or stricter protections (0) following exact policy change dates. Our policy covariate changed from 0 to 1 on 10 October 2005 when Standard Operating Procedure 13.0 (SOP 13) 'Control of Mexican wolves' was implemented by

**Table 2.** Example of monitoring history of a hypothetical wolf ID, broken up into spells for the integration of time-dependent covariates. We use 'analysis time' for the time intervals and order of spells, as covariates change (either *policy* or *season*). The *endpoint* categorical variable is only reflected for the last spell, which corresponds to when monitoring ended (at $t = 250$ in this hypothetical case).

| wolf ID | analysis time when spell begins | analysis time when spell ends | policy | season | endpoint |
|---------|--------------------------------|-------------------------------|--------|--------|----------|
| MX1209  | 0                              | 57                            | 1      | 1      |          |
| MX1209  | 57                             | 140                           | 1      | 0      |          |
| MX1209  | 140                            | 350                           | 0      | 0      | 2        |

the Mexican Wolf Blue Range Reintroduction Project Adaptive Management Oversight Committee (AMOC) and changed back to 0 on 2 December 2009 (table 2). SOP 13 liberalized wolf-killing (1) by establishing a 'three-strikes' policy requiring the permanent removal of wolves implicated in three instances of predation on domestic ungulates during a 1-year period. During SOP 13, removals of wolves more than doubled relative to the previous 7 years [31]. The policy was challenged in court and terminated by a federal judge on 2 December 2009. However, a subsequent change in policy would again liberalize the killing of wolves. Thus, our policy covariate changed from 0 to 1 again from 16 January 2015 to 31 December 2016, which is the last date of our Freedom of Information Act data request (and the end of our study period). On 16 January 2015, the USFWS implemented a modification to the 1998 Endangered Species Act (ESA) 10(j) rule that expands the area where Mexican wolves can be released, allows permitted private entities to kill wolves on non-federal land if wolves are deemed to be a danger to domestic animals and allows killing by government agents on private and state lands if wolves cause unacceptable predation on big game animals.

We also modelled season as a time-dependent covariate using an October–March (winter) and April–September (summer) split, because elsewhere season is known to mediate mortality in wolves [8,19,32,33]. For example, preliminary analysis of a population of Wisconsin wolves revealed winter periods were associated with increases in the hazard and incidence of various endpoints (LTF, poaching, natural death) and at different rates [19]. To model both time-dependent covariates (policy and season), we created splits in each collared wolf's monitoring history. We refer to these splits as 'spells' given they refer to briefer periods within an individual's monitoring time (table 2). In selecting the covariates of interest, we are following best practices of having at least 10 endpoint events per covariate [34–36]. We have, therefore, excluded from our multivariate models any covariates unless they are essential to control.

Nuisance variables are unlikely to confound our analyses as we discuss next. A hypothetical nuisance variable would have to not only correspond to the various changes in policy (2–3) but also be widespread across both NM and AZ across multiple jurisdictions (tribal, state, federal, county lands), and affect multiple independent adult wolves in packs occupying virtually exclusive home ranges. That leaves a climatic event or other widespread biotic event such as a disease with more than one change (to correspond with the policy changes of interest). We have searched both USFWS programme documents and the scientific literature and have found no evidence of changes in environmental events or onsets of disease. Moreover, the covariates that may impact our hypotheses would need to affect the poaching, LTF and legal killing risks. Instead, environmental changes that may covary with the policy may, in fact, show changes to the 'natural' endpoint hazard and incidence, while perhaps affecting the changes in the incidence of our anthropogenic endpoints (through endpoint interactions, see below) but not their hazards.

## 2.2. Statistical methods

We employed endpoint-specific hazard and subhazard models in a competing risk framework, which are extensions of survival (or 'time-to-event') analyses, and a special case of multi-state models [37]. Survival analyses estimate the probability of observing a time interval from the start of monitoring (in our case, release to the wild with a functioning transmitter) to an endpoint, $T$, greater than some stated value $t$, $S(t) = P(T > t)$ within a specified analysis time (our study period above). These techniques allow for calculating the (endpoint-specific) hazard function, $h_k(t)$, or the instantaneous rate of occurrence of a particular endpoint $k$ conditional on not experiencing any endpoint until that time [38–41]. We used the

hazard function to estimate the relative hazard of a collared wolf reaching an endpoint such as LTF, given its survival to a particular date. We used the semi-parametric Cox proportional hazard models to estimate covariate HRs to model how endpoint-specific $h_k(t)$ changes as a function of survival time and model covariates. The estimation of covariate effects on the endpoint-specific hazard is modelled as $h_k(t) = h_{0k}(t)e^{(\beta_1 x_1 + \dots + \beta_j x_j)}$, where $h_{0k}(t)$ is an unestimated baseline hazard function (i.e. semi-parametric) and $\beta_j$ represent the estimates of HRs for each covariate $x_j$ (HR < 1 represents a reduction in hazard and HR > 1 an increase in hazard).

However, hazard rates do not consider competing risks. Competing risk analyses go beyond standard survival analyses by considering multiple endpoints simultaneously (e.g. multiple causes of death, agency removal or LTF). These models are useful for estimating the relative incidence of a particular endpoint, while accounting for other competing endpoints (e.g. the occurrence of human-caused death in the presence of a risk of natural-caused death and LTF). In a competing risk framework, individuals can potentially experience the event of interest (i.e. end of monitoring time) by multiple, mutually exclusive endpoints, although only one is observed. Because the event of interest can only occur due to one endpoint, we refer to the endpoints as 'competing' to bring about that event, and to the respective probabilities over time of that occurring as 'competing risks'.

Competing risk techniques estimate the cumulative incidence (CIF) curve for each endpoint, defined by the failure probability Prob($T < t$, $D = k$); that is, the *cumulative probability* of endpoint $k$ having occurred first (element $D$ is an index variable that specifies *which endpoint* occurred) at time $T$, which specifies *when* the event happened within the study period interval defined over time $t$ *in the presence of other competing endpoints* (i.e. subjects experiencing other endpoints are still considered at risk as individual wolves entered and left the risk set throughout the study period) [37,41,42].

Within the competing risks framework, Fine–Gray (FG) subhazard models estimate differences in CIFs for a given endpoint conditional on covariates [42,43]. FG models use regression techniques similar to the Cox model, except parameter interpretation changes as follows: estimates are interpreted as SHRs or relative incidence (rather than HRs) in the presence of other endpoints (i.e. for each covariate $x_j$: SHR < 1 represents a reduction in incidence and SHR > 1 an increase in incidence). Although both hazard and competing risk models are informative, the competing risk models consider more information and provide greater predictive power [41,42,44].

Hence, while endpoint-specific Cox models and their HRs allowed us to test the hypothesis that liberalized wolf-killing affected the *rate of occurrence* (i.e. hazard) of any endpoint relative to policy periods, the FG models and their SHRs allowed us to test if and how much liberalized killing affected the *probability* and *incidence* of endpoints, in addition to the potential simultaneous effects of other covariates. CIFs allowed us to visualize those effects on incidence while considering the prevalence of each endpoint in the population. Therefore, we used both hazard and incidence to infer the changes due to policy period and test the opposed hypotheses (table 1).

Our Cox proportional hazards and FG subhazard models comply with the appropriate number of events per variable recommended in the scientific literature to ensure the accurate estimation of regression coefficients and their associated quantities for the endpoints of interest (poached, agency removals, LTF) (see tables 3 and 4) [34,35,45].

Following recommendations for rigorous competing risk analysis [41,42,44,46], we reported results on all endpoint-specific hazards and CIFs to elucidate how hazards and incidences of multiple endpoints interact. For example, analysis of Wisconsin wolves suggested the increases in both the hazard and incidence of LTF during liberalized killing periods offset and potentially overcompensated for the smaller decreases in hazard and incidence of reported wolf-poaching estimated during those same periods [19].

Finally, we used Bayes' factor (BF) [47] to assess the strength of evidence for each of our alternative hypotheses and the null hypothesis for each poaching endpoint. We used the free BF online calculator found at http://www.lifesci.sussex.ac.uk/home/Zoltan_Dienes/inference/Bayes.htm, which assumes parameter estimates are normally distributed with known variance. The parameters used for each endpoint will be its point estimate and s.e. derived from the final Cox and FG models for testing HRs and SHRs, respectively. To assess the robustness of our conclusion to our prediction of the population distribution given our hypotheses and because prior theoretical support for any particular BF specification is scant, we assumed three different likelihood functions for the hypotheses' predicted effect as recommended by Dienes [47]: (i) a half-normal function using the legal removal endpoint's point estimates to model the expected standard deviation as s.d. = point estimate, (ii) a uniform function using the legal removal endpoint for Mexican wolves as the upper bound and 0 as the lower bound, and (iii) a half-normal using endpoint-specific estimates from [19] to calculate the s.d. in the same manner as (i) [47,48]. In doing so, we follow Dienes' [47] recommendations to use likely values while keeping our

**Table 3.** Number of endpoints (unique wolf IDs) during periods of liberalized killing or periods of stricter protections for step 1 (diagnostic step). Wolves that survived to the end of the study period ($n = 52$) are omitted here and censored in analyses. The study period spanned 29 March 1998 to 31 December 2016 inclusive.

| endpoint | stricter protection, policy period = 0 ($t = 4621$ days) | liberalized killing policy period = 1 ($t = 2230$ days) |
|---|---|---|
| agency removal | 28 | 20 |
| natural cause | 11 | 11 |
| human cause | 55 | 35 |
| LTF | 27 | 40 |

**Table 4.** Number of endpoints (unique wolf IDs) during periods of liberalized killing or periods of stricter protections for step 2 using OLE data from investigations of suspicious deaths. Wolves that survived to the end of the study period ($n = 52$) are omitted here and censored in analyses. The study period spanned 29 March 1998 to 31 December 2016 inclusive.

| endpoint | stricter protection, policy period = 0 ($t = 4621$ days) | liberalized killing policy period = 1 ($t = 2230$ days) |
|---|---|---|
| agency removal | 28 | 20 |
| poaching | 35 | 17 |
| natural death | 11 | 11 |
| non-criminal | 20 | 18 |
| LTF | 27 | 40 |

predictions blind to the data as required by a registered report. In the absence of a base of prior literature to guide us, our use of these default variances maintains the required 'blind' to our data with a reasonable estimate of variability in the distribution around the point estimate. We use the legal removal endpoint estimates (rather than other imperfectly reported endpoints) in two specifications of our hypotheses because we know there is an effect (i.e. more wolves are killed legally during legalized killing periods). We report BFs for all HRs and SHRs of interest (i.e. reported poached and LTF). BFs strength of evidence for each hypothesis (or null) was interpreted as follows: $1/3 < BF < 3$ would be inconclusive evidence; $BF > 3$ would be substantial evidence for the alternative hypothesis; $BF < 1/3$ would be substantial evidence for the null hypothesis (i.e. no effect) [47,48]. Given our three BF specifications for each endpoint parameter, we would conclude in favour of a particular hypothesis if most BF specifications (two out of three BFs) support that conclusion. Accordingly, we might generate contradictory evidence (strong support for each hypothesis and the null) or inconclusive evidence (failure of any hypothesis to survive a majority of the BF specifications). For purposes of comparison, and to provide estimates of policy effects on 'total potential' (cryptic + reported) poached, we also aggregate and run all analysis on the new endpoint LTF + POA (including the procedure for pre-specified BFs as above).

## 2.3. Diagnostic step

Because we used information from two data sources (MWRP and OLE datasets), we analysed the data in two steps to provide more nuance about the effect of the policy intervention on mortality and disappearance. Both steps employed all survival and competing risk analyses previously described. We analysed four endpoints: 'human', 'natural', 'LTF' and 'agency removal' (table 3). The drawback from this endpoint breakdown is the inability to conclude anything directly regarding any policy effects on subsets of anthropogenic mortality (e.g. poaching or non-criminal human-caused deaths).

## 2.4. Analyses

We added OLE data on human-caused endpoints and further specified which were poaching and which were deemed non-criminal (table 4). However, this comes at the expense of lower number of observations in each human-caused endpoint and reduced statistical power (fewer events per variable, see the

previous section and tables 3 and 4). Thus, results from the diagnostic step above will prove more statistically robust, but the exploration of the various anthropogenic endpoints is imperative, given evidence of different policy effects on each [19].

By evaluating the effect(s) of both liberalized killing periods (SOP 13 and revised 10(j) rule), we will strengthen the inference about the policy intervention with a better case–control design (reverse-treatment or before–during–after–during). Therefore, both policy periods will be sampled twice.

Two divisions of the USFWS did the preliminary quality check on data. First, the Mexican Wolf Recovery team collected mortality and disappearance data with a simple endpoint classification as 'died of natural cause, died of human cause, legal removal by agency, disappeared (LTF)'. Next, for the subset of human-caused deaths above, the independent USFWS OLE classified deaths by cause (vehicle, poaching, accidental) and occasionally reclassified a human-caused death as natural after detailed investigations, some or all of which included necropsy, radiography or field investigation. Within our team, the two co-lead authors performed interobserver reliability tests by independently taking the data provided by the USFWS and summarizing it (tables 3 and 4), then comparing and resolving discrepancies (arbitrated by the senior author in the case of disagreements).

In Phase 2, the two co-lead authors separately prepared the data for analysis by aligning individual wolf survival histories with covariates and the intervention (policy period). The senior author A.T. is F.J.S.-Á.'s post-doc supervisor and N.X.L.'s PhD advisor. He and D.R.P. served as a sceptical trouble-shooter for analyses, interpretation and writing—blind to results until the Phase 2 analysis was considered complete by the two co-lead authors. This team organization and separation of powers was intended to reduce bias and improve the strength of inference.

# 3. Results

The two policy periods we examined which liberalized killing of Mexican wolves resulted in various changes in the hazard and incidence of endpoints of collared wolves relative to the two periods of stricter protection. The diagnostic step (electronic supplementary material, S1) and results that follow confirmed the importance of disaggregating human-caused endpoints to assess the evidence for our alternative hypotheses.

Covariates of winter and sex did not significantly affect the results of any models, and, therefore, the most parsimonious model included the policy intervention without either covariate. The proportional hazard assumption of Cox models was met for all endpoints (see electronic supplementary material, figures S7 and S8). For information about each model and their parameters, see electronic supplementary material, table S1. The best models revealed the following changes for collared Mexican wolves.

## 3.1. Lost to follow-up

Periods of liberalized killing were associated with a 121% increase in the hazard for the endpoint of LTF, relative to periods of stricter protection, compatible with a positive range (does not include zero) of +36% to +260% (HR = 2.21, $p < 0.001$; table 5 and figure 1$b$). The proportion of collared wolves (CIF) with the endpoint of LTF increased by 128% (SHR = 2.28, compatible interval = 38–276%, $p < 0.001$) during periods of liberalized killing relative to periods of stricter protection (table 6 and figure 1$a$).

## 3.2. Reported poached

Periods of liberalized killing were associated with a 22% decrease in hazard for the endpoint of reported poached, relative to periods of stricter protection, compatible with a range that overlaps zero of −56% to +39% ($p = 0.407$; table 5 and figure 1$d$). The proportion of collared wolves (CIF) with the endpoint of reported poached decreased by 31% (SHR = 0.69, compatible interval = −62% to +25%, $p = 0.226$) during periods of liberalized killing relative to periods of stricter protection (table 6 and figure 1$a$).

## 3.3. Agency removal

Periods of liberalized killing were associated with a 5% increase in hazard for the endpoint of agency removal, relative to periods of stricter protection (HR = 1.05, compatible interval includes zero, −41% to +88%, $p = 0.863$, table 5 and figure 1$c$). The proportion of collared wolves (CIF) with endpoint of agency removal decreased by 4% (SHR = 0.959, compatible interval includes zero, −48%

**Table 5.** Cox model of cause-specific hazards for each endpoint for 279 collared Mexican wolves. HR < 1 represents a reduction in hazard during periods of liberalized killing (lib_kill = 1) and HR > 1 denotes an increase in hazard, HR < 1 a decrease and HR = 1 no change in hazard. Only the most parsimonious model is presented (see electronic supplementary material for all models). In all cases, the proportional hazard assumption of the Cox models was met. Comp. Int., compatible interval around point estimates.

| variable | endpoint | | | | | | | | | |
| | lost to follow-up (LTF) | | agency removal | | reported poached | | non-criminal | | natural | |
| | HR | Comp. Int. | HR | Comp. Int. | HR | Comp. Int. | HR | Comp. Int. | HR | Comp. Int. |
|---|---|---|---|---|---|---|---|---|---|---|
| liberalized killing periods (lib_kill) | 2.21* | 1.36–3.60 | 1.05 | 0.59–1.88 | 0.78 | 0.44–1.39 | 1.42 | 0.75–2.71 | 1.28 | 0.53–3.08 |

*$p < 0.001$, all other results had $p > 0.05$.

**Table 6.** FG competing risk models of cause-specific subhazard (SHR) for each endpoint for 279 collared Mexican wolves. SHR < 1 represents a reduction in the incidence of the endpoint during periods of liberalized killing (lib_kill = 1) and SHR > 1 an increase in incidence. SHR = 1 would represent no change in relative incidence. Only the most parsimonious model is presented (see electronic supplementary material). Comp. Int., compatible intervals around point estimates.

| variable | endpoint | | | | | | | | | |
| | lost to follow-up (LTF) | | agency removal | | reported poached | | non-criminal | | natural | |
| | SHR | Comp. Int. | SHR | Comp. Int. | SHR | Comp. Int. | SHR | Comp. Int. | SHR | Comp. Int. |
|---|---|---|---|---|---|---|---|---|---|---|
| liberalized killing periods (lib_kill) | 2.28* | 1.38–3.76 | 0.96 | 0.53–1.75 | 0.69 | 0.38–1.25 | 1.27 | 0.66–2.46 | 1.32 | 0.56–3.11 |

*$p$-value $< 0.001$.

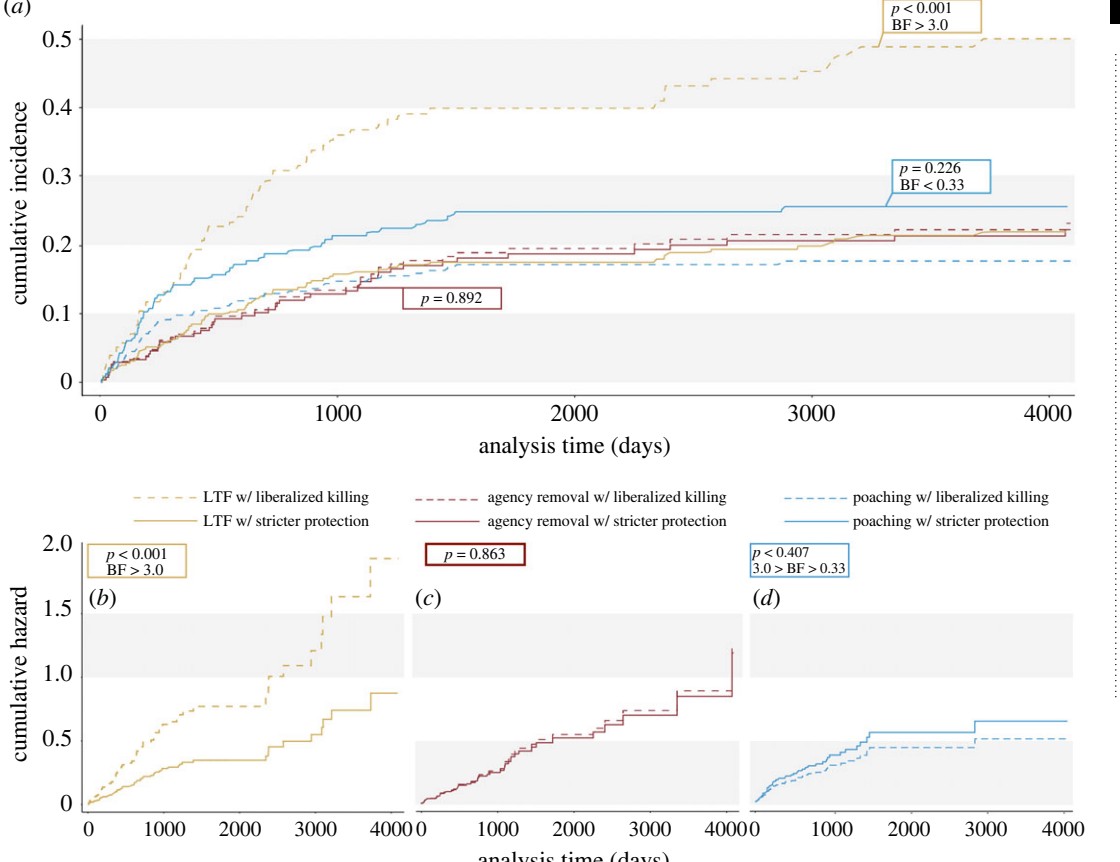

**Figure 1.** Cumulative incidence functions (CIF) (*a*) derived from FG subhazard models and hazard functions derived from univariate Cox models (*b–d*) for 279 collared Mexican wolves during periods with reduced protections for wolves (liberalized killing periods = 1, dashed lines) and periods with stricter protections (solid lines) for three independent endpoints (LTF, *n* = 67; reported poached, *n* = 52; and agency removal, *n* = 48). See electronic supplementary material for the cause-specific hazard functions (electronic supplementary material, figures S1 and S2) and CIFs (electronic supplementary material, figures S3 and S4) for the natural and non-criminal endpoints. BF supports a hypothesis over the null when greater than 3, supports the null over a hypothesis when less than 0.33 and represents inconclusive evidence for either hypothesis with 0.33 < BF < 3 [47]. (*a*) CIF curves show the proportion of collared wolves disappearing (LTF, yellow line) was significantly greater than other endpoints, during periods of liberalized killing (SHR = 2.28, compatible interval = +0.38 to +2.76, *p* < 0.001). (*b–d*) Lines show cumulative hazard over analysis time (days of monitoring each wolf). (*b*) LTF HR = 2.21 (compatible interval = +0.36 to +2.60). (*c*) Agency removal HR = 1.05 (compatible interval = −0.41 to +0.88). (*d*) Reported poached HR = 0.78 (compatible interval = −0.56 to +0.39).

to +75%, *p* = 0.892) during periods of liberalized killing relative to periods of stricter protection (table 6 and figure 1*a*).

## 3.4. Non-criminal

Periods of liberalized killing were associated with a 42% increase in hazard for the endpoint of non-criminal causes, relative to periods of stricter protection (HR = 1.42, compatible interval includes zero, −25% to +171%, *p* = 0.278, table 5). The proportion of collared wolves (CIF) with the endpoint of non-criminal increased by 27% (SHR = 1.27, compatible interval includes zero, −33.7% to +146%, *p* = 0.463) during periods of liberalized killing relative to periods of stricter protection (table 6).

## 3.5. Natural

Periods of liberalized killing were associated with a 28% increase in hazard for the endpoint of natural causes relative to periods of stricter wolf protections (HR = 1.28, compatible interval includes zero, −47%

to +208%, $p = 0.582$, table 5). The proportion of collared wolves (CIF) with the endpoint of natural increased by 32%, compatible interval = −44% to +211% (SHR = 1.32, $p = 0.526$; table 6).

## 3.6. Total potential poached

Periods of liberalized killing were associated with a 42% increase in hazard for the endpoint of total potential poached (aggregated endpoint of LTF and reported poached), relative to periods of stricter protections (HR = 1.42, compatible interval = −0.7% to +103%, $p = 0.05$). The proportion of collared wolves (CIF) with endpoint of total potential poached increased by 38% (SHR = 1.38, compatible interval = −5% to +101%, $p = 0.095$).

## 3.7. Bayes factors

We calculated BF with three specifications to model the expected predicted and maximum effect (electronic supplementary material, table S2). BFs were inconclusive (1 < BF < 3) for most endpoints, suggesting the data are inconclusive for distinguishing the hypotheses from the null, following recommended criteria for BF interpretation (table 1). Indeed, all three BF specifications proved inconclusive for the reported poached and total potential poached endpoints, which does not support the killing for tolerance hypothesis. However, our presumption that agency removal increased during periods of liberalized wolf-killing was not supported (HR = 1.05 and SHR = 0.96); therefore, specifications 1 and 2 seem meaningless (table 7). By contrast, the third specification of BF based on endpoint-specific estimates of predicted effect as recommended by Dienes [47,48] were meaningful (table 7). These are conclusive evidence of an increase in LTF during liberalized wolf-killing periods (BF = 8.08); that specification was inconclusive on changes for reported and total potential poached endpoints. See electronic supplementary materials for information about the inputs used to calculate BFs (electronic supplementary material, table S2).

# 4. Discussion

Here, we report a replication of the findings of Santiago-Ávila et al. [19] that grey wolves that disappeared from monitoring did so at higher rates during periods of reduced protections (i.e. liberalized killing) than during periods of stricter protections under the US Endangered Species Act (ESA). We find stronger evidence for this pattern among collared Mexican wolves (C. l. baileyi) than was found among collared grey wolves in Wisconsin, USA [19]. Because the disappearance of collared wolves that are being monitored by VHF or GPS is most often caused by illegal activities [5,7,49], the present study further undermines the common assumption that animals lost to monitoring suffer from the same hazards and endpoints as those animals that are perfectly monitored [4].

In the paragraphs below, we first justify the assertion that the observed pattern in disappearances results from increases in cryptic poaching. Second, we conclude that our results support the 'facilitated illegal killing' hypothesis and do not support the 'killing for tolerance' hypothesis. Third, we discuss strength of inference in this subfield of wildlife science including protections against bias used in this study for the first time in this subfield to strengthen inference. Fourth, we propose that legal killing, non-lethal removal from the wild, and facilitation of cryptic poaching are all impediments to endangered wolf recovery under the ESA. Finally, we discuss the general lessons we draw from this study for the US federal agency implementing the ESA (USFWS) for wolves generally, for other countries and for anti-poaching research and intervention.

The relative stability in hazard and incidence of all known fates (not including the subset of radio-collared Mexican wolves that were designated LTF, lost to follow-up) between policy periods would suggest that if LTF wolves were in fact lost due to the same endpoints as monitored individuals with known fates, then we should observe relative stability in hazard and incidence of LTF between policy periods. This is not the case in either this study or [19]. Past work provides numerous independent lines of evidence that the majority of LTF could not be emigrants nor transmitter failures. First and most importantly, in both this study and that of Santiago-Ávila et al. [19], hazard and incidence rates of LTF changed in correlation with policies on legal killing, which could not plausibly have caused transmitter failures; see also [49] on different rates of LTF between hunting and non-hunting seasons. Also, battery life might be confounding LTF that occurred long after collaring. Contrary to this expectation, LTF was much shorter (average 788 days) than the average length of time to the natural mortality endpoint (1175

**Table 7.** BF calculations for reported poached. LTF and aggregated 'total potential poached' (LTF + POA) endpoints for collared Mexican wolves using three specifications: (i) a half-normal distribution using the Mexican wolf agency removal endpoint point estimate of HR and SHR; (ii) a uniform function using the agency removal endpoint for Mexican wolves as the upper bound and 0 as the lower bound, and (iii) a half-normal distribution and the analogous estimates of HR and SHR from Santiago-Ávila et al. [19]; see electronic supplementary material for all parameters. BFs strength of evidence for each hypothesis (or null) was interpreted as follows: $1/3 < BF < 3$ (ref) would be inconclusive evidence; $BF > 3$ would represent substantial evidence for the alternative hypothesis; $BF < 1/3$ would represent substantial evidence for the null hypothesis of no association.

| | endpoint | | | | | |
| | LTF | | POA | | LTF + POA | |
| BF specifications | HR | SHR | HR | SHR | HR | SHR |
| --- | --- | --- | --- | --- | --- | --- |
| (i) half-normal w/MX-agency removal | 1.8 | 0.69 | 0.89 | 1.14 | 1.15 | 0.76 |
| (ii) uniform w/upbound-MX-agency removal | 1.41 | 1.30 | 0.93 | 0.92 | 1.09 | 1.19 |
| (iii) half-normal w/WI POA | 8.08 | 8.08 | 1.25 | 1.63 | 1.35 | 2.02 |

days). If battery life were the confounding factor, we would expect the average time to LTF to be more similar to the average time to the natural mortality endpoint. Second, if LTF were largely made up of emigrants from the Mexican wolf recovery area, some of these individuals would probably have been found dead in surrounding areas by citizens with nothing to hide who presumably would have reported their observations to authorities. The USFWS databases we used contained no such cases. Therefore, LTF wolves were most likely killed and the evidence of the illegal action was concealed, e.g. by destruction of transmitters. Such cryptic poaching was first estimated by Liberg et al. [7], and subsequently explored in Treves et al. [4,5] and Santiago-Ávila et al. [19], and could have been exacerbated in the Mexican wolf recovery area by the policy of sharing radiofrequencies of collared wolves with members of the public [50]. We conclude that our results on the disappearances of collared Mexican wolves reinforce those first reported by Treves et al. [5] which demonstrated the bias introduced by excluding disappearances of marked wolves in mortality analyses for four endangered wolf populations.

We tested opposed hypotheses about the effect of legalizing killing or removal of individuals of an endangered species on the survival of collared individuals remaining in the wild. The USFWS, responsible for implementing the ESA for terrestrial species, has been a particular supporter of the 'killing for tolerance' hypothesis. It has repeatedly invoked this hypothesis in its endangered grey wolf management under the assumption that government-permitted killing of grey wolves would mitigate or prevent illegal killing and raise public tolerance for wolves, so ongoing recoveries would not be stopped or slowed by illicit resistance [9,51]. A federal court rejected that argument as a speculative approach to abridging the ESA prohibitions that are explicitly aimed at preventing killing [9] but in 2017, another court seemed to defer to the agency on this point when it wrote '… it is clear that in drafting the present Section 10(j) rule, the take provisions are critical to conciliating those opposed to the reintroduction effort,…' [52, p. 43]. Despite this deference, the court remanded the rule to the USFWS to repair its deficiencies. We recommend the USFWS abandon the expectation and repudiate the oft-repeated and unsupported notion that liberalizing killing would reconcile opponents of wolf protection. The results of this study support the alternative and mutually exclusive hypothesis that liberalized killing policies *facilitate illegal killing* and join a growing body of evidence that suggests liberalized killing policies lower tolerance for wolves and slow wolf population growth substantially more than expected from legal killing rates.

This study, therefore, adds to the literature regarding policy effects on anthropogenic causes of wolf mortality. Human dimensions research using focus groups and mail-back surveys measuring attitudes of Wisconsin residents in and out of wolf range found that respondents' tolerance for wolves decreased and reported respondents' willingness to poach wolves increased after wolf-killing was liberalized seven times between 2003 and 2013 [12,14]. Moreover, calls for more killing of wolves followed relaxing ESA protections [13]. Further, a study of population dynamics showed with 92% certainty that there had been an unexplained decrease in the growth of wolf populations in both Wisconsin and Michigan after wolf-killing was liberalized, independent of the number of wolves killed legally [15,16]. The latter authors could only explain the repeated, parallel slow-downs by the existence of undetected mortality

[15,16], a result which withstood a series of attempts at rebuttal that did not include new data [17] and in one case muddied the waters with errors and omissions [23]. The conclusion from Chapron & Treves [15,16] that the length of the policy period was predictive of the population growth slow-down independent of the reported number of wolves killed is consistent with our finding that exposing Mexican wolves to liberalized killing was associated with higher hazard and incidence of LTF, not predicted by the hazard or incidence of wolves legally killed. Therefore, three independent lines of evidence point in the same direction and opposite to the USFWS hypothesis about liberalizing killing.

Moreover, here we provide more direct, stronger inference than ever before, against the government's 'killing for tolerance' hypothesis in a new population of wolves. Given that periods of reduced protections allowed for greater flexibility in legal lethal actions towards Mexican wolves, we expected to observe a higher increase in the hazard of agency removals during periods of reduced protections (HR = 1.05). Therefore, we reject the possible explanation that increased hazard or incidence of agency removal (SHR = 0.95) during liberalized killing periods is somehow leading to an increase in disappearances (such as emigration or super-additive mortality). Rather, it was the policy period announcement or its duration *per se* that had the effect of increasing collared wolf disappearances. Chapron & Treves [15] proposed that reducing protections for wolves sends a policy signal lowering the value of wolves to the public including would-be poachers or reducing the likelihood of enforcement against poaching. The hypothesized 'policy signal' seems to convey that either the lives of individual wolves are perceived as less valuable, the benefit of wolves has declined, or prosecution of poachers will relax. We find little evidence to support the latter, because the LTF endpoint represents destruction of evidence of poaching. Instead, would-be poachers appeared to have opted to act cryptically or increase their concealment of evidence during periods of reduced protection, an inference that is supported by a recent news report [53]. The inference that would-be poachers became more concerned with law enforcement while increasing their poaching rates is consistent with the current study in Mexican wolf range and that of Santiago-Ávila *et al.* [19] for Wisconsin wolves.

Until sophisticated, replicable studies of confirmed poachers and their attitudes are conducted, we cannot know if would-be poachers responded to policy signals by repeating past poaching behaviour with the addition of more concealment of evidence, or if new actors began poaching with concealment. We predict a mix of both patterns, but a preponderance of the poaching during periods of liberalized killing was by individuals who now chose to conceal evidence. That pattern would be supported if the USFWS began to give out radiofrequencies of collared wolves or otherwise changed agency conduct in the field in such a way as to expose collared wolves to higher risk.

In past studies, including those from the US Midwest, as well as one performed in Scandinavia [8], periods of reduced wolf protection were associated with significant increases in hunting or government lethal removal of wolves. Rates of wolf disappearances or poached wolves also increased, but not as drastically as we observe here. These less drastic changes for other endpoints may be a result of 'cleaning up the numbers' [8,19]; more wolves reach the endpoint of legal killing before they can succumb to some other endpoint, such as reported or cryptic poaching, thereby muddying our understanding of the effect of legal killing policies on other fates of collared wolves. Our study is not confounded by any effect of 'cleaning up the numbers' because Mexican wolves were not subject to higher hazard of legal removal and the incidence of wolves lost by agency removal decreased, yet radio-collared wolves disappeared at higher rates. Rather, we detected significantly more hazardous conditions for the critically endangered Mexican wolf when the USFWS reduced ESA protections independent of agency removal.

Regarding the unexpected finding that the rate of agency removal changed little as policy periods changed from stricter protection to reduced protection, we re-examined our starting assumption. We based our starting assumption of a higher increase in agency removals during two periods of reduced protection (SOP 13 and revised 10(j) rule) on two pieces of likely misleading information. First, by examining the raw numbers of agency removals in tables 3 and 4, it appears that the prevalence of wolves being removed during periods of liberalized killing is about 150% of that being removed during periods of stricter protections. However, hazard and incidence, as we calculate here with survival analysis methods, are based off the sum of all the days each wolf was exposed to each endpoint (i.e. their aggregate time-at-risk). Therefore, time-at-risk is much greater than the number of days over particular policy periods (as reported in tables 3 and 4). We further based our expectations of the change in agency removals over policy periods on a source which claimed that agency removals more than doubled during the SOP 13 policy relative to prior years (see Methods) [31]. However, Fitzgerald [31] lacks any accompanying data.

An often-overlooked aspect of wolf mortality reporting is how the agency classifies cause of death when human-caused. Treves *et al.* [4] mentioned consolidating causes of death such as non-permitted trapping, shooting, poisoning into one category of poaching, especially when the agency might otherwise misidentify the primary cause of death. We observed a pattern in the data for Mexican wolves that was not detected in Santiago-Ávila *et al.* [19]. In particular, the incidence of the non-criminal endpoint increased by 27% during periods of reduced wolf protections (table 6). Our results (HR = 1.42 and SHR = 1.27) for the non-criminal endpoint suggest the possibility that USFWS staff classified a greater number of anthropogenic causes of death as non-criminal during periods of reduced protections. This may be a logical result of liberalizing killing, as less killing is legally classified as 'poaching'. Indeed, some poaching could merely have been reclassified as non-criminal by USFWS staff using subjective definitions and thereby confirming the (erroneous) perception that poaching had diminished because of 'increased tolerance'. However, the non-criminal endpoint includes vehicle collisions, and other human-caused mortalities that were classified as non-criminal after an investigation by the USFWS Office of Law Enforcement (OLE). Therefore, it is impossible to know the real cause of the increase in hazard and incidence of the non-criminal endpoint during periods of reduced protections without knowing more about the OLE investigations. We received two datasets for Mexican wolves from the USFWS. One set pooled all anthropogenic mortality in one category, which is clearly less useful for analyses such as ours that can tease apart the effect of policy interventions on specific endpoints. Hence, the second dataset from OLE which assigned criminal and non-criminal causes of death and also distinguished further subcategories was much more useful to us. We surmise it would have been more useful to managers and the public also. Therefore, we recommend the USFWS share data on mortality of endangered species that are disaggregated into no fewer than four categories (legal, illegal, vehicle collision and natural), report disappearances (LTF) systematically in the same tables along with start and end dates for time on the air. By the same token, too many categories of poaching as a cause of death can obscure the priority of illegal killing.

Some readers might wonder if frustration among would-be poachers rose in the Mexican wolf range because agency removals did not change between periods, despite the policy signal that protections were loosened. However, we argue that if this were a valid explanation, we would expect frustration with the lack of change in the rate of lethal actions to be prevalent during both policy periods, and we would anticipate the rate of disappearance of wolves to be comparable during the two periods, which we do not see. We would, therefore, expect a different pattern than in the US Midwest where the agency did use liberalized killing periods to lethally remove wolves at higher and increasing rates [19]. Proponents of the frustration hypothesis claimed Wisconsin's would-be poachers were frustrated when protections were tightened [11]. No plausible cognitive mechanism that would differ between would-be poachers of Wisconsin and those of New Mexico/Arizona has been presented. The frustration hypothesis requires that two different cognitive behavioural mechanisms exist in the two populations, which does not seem parsimonious and is not consistent with the attitudinal data from Wisconsin (see above). This Mexican wolf study cannot support the USFWS idea that without legal recourse, actors would take matters into their own hands, because would-be poachers observing agency removals would be expected to be as frustrated in and out of the policy periods examined here.

Because this study and Santiago-Ávila *et al.* [19] used time-series analysis for before-and-after comparison of interventions (BACI) without randomization to control or treatment, it provided stronger inference than prior work that relied on correlation and single point estimates [11,15,16], by accepted standards from other fields [54,55]. The standards have been explained at length in [56–58] in relation to their application to evaluation of methods to prevent carnivore attacks on livestock. The current study also integrated several novel protections against bias that further strengthen the inference. An additional reduction in confounding variables in this study is that just over half of the wild Mexican wolves were marked and monitored, compared to the average of 13% marked in the Wisconsin population. That increases the strength of generalizations about Mexican wolves as a whole and increases our confidence in parameter estimates for known and unknown fates in the present study. Therefore, only evidence using experimental controls could achieve stronger inference than does the current study.

Furthermore, this study directly attempted to reduce bias in the following ways. First, we used official data as classified by the management agencies in charge (both USFWS MWRP and its OLE), not by us. Therefore, if any bias exists in the classification scheme, it reflects classification decisions by the agencies and probably random error given the agencies were apparently unaware of the hypotheses. Second, by publishing methods before completing an analysis, we ensured that the analyses could not be amended to support one or another hypothesis. Further, peer-review of the methods helped our team develop stronger analysis methods that would allow us to better interpret the strength of the evidence. Fourth,

we used BF rather than arbitrary traditional significance thresholds to assess the relative strength of evidence for our hypotheses. Our team also developed internal safeguards by having separate members of the team independently interpret the results without taking part in analysis. The ESA requires the use of best available scientific and commercial data, hence policymakers wishing to implement the ESA as intended by Congress can take comfort that the science has advanced to the highest level, rather than continuing to debate with imperfect evidence as the scientific community did from 2013 to 2019 (reviewed in the section on hypothesis tests).

We conducted a BF analysis to quantitatively assess the strength of evidence for our two competing hypotheses and the null hypothesis. Calculating BFs for each endpoint allows us to go beyond significant or non-significant results to examine whether non-statistically significant results truly represented evidence against either hypothesis [47]. To determine whether our results represent evidence for or against either opposing hypothesis or the null hypothesis, Dienes [47] recommends using prior published research to determine what our theory predicts. The theories we test here have not been widely tested; therefore, our best source for endpoint-specific parameter estimates came from Santiago-Ávila et al. [19].

Those prior results from an unrelated dataset provided a comparison for the Mexican wolf results in a registered report but prior to analysing the data. We calculated the BFs of our results using three specifications (defined in Methods) and, following our stated interpretation criteria for BFs detailed in the Methods and table 1, none of our endpoints of interest (LTF, poached, total poached) provided conclusive evidence for either of our alternative hypotheses (electronic supplementary material, table S2). However, we have greater confidence in the results calculated using the third specification; the half-normal distribution calculated using prior data; Santiago-Ávila et al.'s [19] HR and SHR values for the LTF and 'reported poached' endpoint. Our rationale for our confidence in the third specification is as follows: (i) the effects observed in both studies are endpoint-specific; therefore, the estimates used ([19]'s HR and SHR) are a result of similar mechanisms, rather than using estimates from different endpoints that probably result from different mechanisms, as in our other two specifications, both based on using the Mexican wolf agency removal endpoint for comparison; (ii) when submitting our methods as a registered report, we had to make a 'blind' (prior to analyses) assumption regarding the change in agency removal endpoint for Mexican wolves (i.e. that agency removals would increase with liberalized protections), which proved counterintuitively unchanged between policy periods, thereby eliminating its potential predictive power; and (iii) the agency removal and reported poached HRs are opposite in direction and the same occurs with the agency removal and LTF SHRs, so those estimates do not provide plausible parameters for the reported poached and LTF endpoints.

The only BF that was conclusive was the support for the 'facilitated illegal killing' hypothesis shown by the increase in disappearances of collared Mexican wolves (LTF, table 7). By contrast, the corresponding BF for the total potential poached endpoint fell below the criterion level, because the aggregated LTF and reported poached endpoints ran in opposite directions, but the increase in the proportion of wolves with fate LTF was nearly three times greater than the decrease in the proportion of wolves reported poached. None of the BF analyses supported the killing for tolerance hypothesis (table 7). We conclude the USFWS claim in federal court that lessening ESA protections with the 10(j) rule would reconcile opponents of reintroduction and in turn be harmless for the Mexican wolves in the wild [59] now seems untenable.

## 4.1. Implications for endangered species

Policy interventions should be effective, i.e. achieve their goals, without serious, unwanted side effects. This study finds that for Mexican wolves, there were serious side effects of the liberalized killing policies. The increase in disappearances of Mexican wolves we detected was substantial during those periods of reduced protections, despite a lack of change in the rate of government removal of wolves. Unplanned, unregulated disappearances are wasteful: a waste of taxpayer money spent on telemetry and relocation to the wild; a loss of individual animals that are unique and irretrievable by known technology; a waste of private resources used in their captive breeding; and undermines the role of the federal government as trustee of US wildlife since 1842 [3]. The effect we found also demonstrates widespread unlawful disregard for the most popular environmental law ever passed in the USA [60].

Further, the policy of liberalizing killing cannot be justified by the vague and indirect claim that it speeds population growth at the expense of individual survival because USFWS data show that the Mexican wolf population declined from 55 to 42 during a 6-year period of liberalized killing from 2004 to 2009 [50]. Similarly, after the implementation of the 10(j) revised rule in 2015, the Mexican wolf population in the wild declined 12%, partially rebounded the next year, and did not change by

2017 when the court order remanded the revised 10(j) rule to the USFWS. Thereafter, growth continued at the prior rate averaging 22% per year (fig. 5 in [50]). The latter finding replicates that of Chapron & Treves [15–17,23] for Wisconsin's and Michigan's grey wolves. Currently, the Mexican wolf population numbers 163 [61]. In view of these results, we hypothesize that population growth will slow, halt and maybe even reverse, given currently authorized liberalized killing, with the effect on growth mediated by the magnitude of the policy signal on disappearances; that is, on cryptic poaching rather than agency removals. However, protections for Mexican grey wolves could be strengthened in a revised 10(j) rule being considered by USFWS.

In this context, the balance tilts towards the Mexican wolves by law (quoting the court in Center for Biological Diversity *v.* Jewell (2018) 'Harm to endangered or threatened species is considered irreparable harm, and the balance of hardships will generally tip in favor of the species. *See Marbled Murrelet v. Babbitt*, 83 F.3d 1068, 1073 (9th Cir. 1996) ("Congress has determined that under the ESA the balance of hardships always tips sharply in favour of endangered or threatened species."); *Amoco Prod. Co. v. Vill. of Gambell*, *AK*, 480 U.S. 531, 545 (1987) ("Environmental injury, by its nature, can seldom be adequately remedied by money damages and is often permanent or at least of long duration, i.e. irreparable. If such injury is sufficiently likely, therefore, the balance of harms will usually favor the issuance of an injunction to protect the environment.")' p. 23, Docket CV-15-00019-TUC-JGZ, U.S. District Court Arizona, 2018).

The issue goes beyond Mexican wolves. As recently as 14 December 2020, the USFWS continued to espouse the unsupported view that reducing or removing ESA protections will help individual wolves to survive and help wolf populations to recover [51]. In the latter letter to the State of California Fish & Game Commission, the USFWS cited outdated studies that have been superseded since 2016, with frequent communications [62–64].

There are lessons in the current work that have implications beyond the USA and beyond wolves. We suggest other national policies for killing large predators (or other non-humans) to raise tolerance or lower poaching should be scrutinized for strength of inference and the quality of evidence (e.g. [21,22]). The notion that protection for large carnivores generates poaching as a form of rural resistance has merit, but the suggestion that relaxing protections is the solution is no longer credible (contra Kalternborn & Brainerd [65]). Similarly, the long-held notion that without compensation for losses, affected people will kill wildlife illegally, needs re-examination in the light of our current results. The alternative is that compensation might encourage hostage-taking, i.e. an escalation by affected parties to threatening endangered species if they are not better compensated for property damages. Our findings join an active debate about leniency versus enforcement as more functionally effective conservation interventions.

In cases of wildlife trade, such as ivory or rhino horn, arguments for leniency are focused around creating a legal trade to inhibit the lucrative illegal trade [66,67]. However, sources of wildlife crime stemming from conflict, such as in the case of grey wolves, may not be effectively understood, nor managed in the same ways. Leniency has been tried for grey wolves, and evidence suggests leniency fails to achieve greater tolerance and reduced wildlife crime. On the contrary, leniency is associated with increased poaching of wolves in the USA. We encourage the scientific evaluation of all candidate interventions as experiments, preferably with suitable comparisons or even experimental controls, with safeguards against bias.

We suspect wolves are not exceptional among large carnivores regarding the effect of relaxing or fortifying legal protections, because the same justifications for liberalizing killing of brown bears and lions have been used whenever prohibitions on hunting or other lethal management are proposed [10,68–70]. We urge that similar studies be completed to examine whether there is in fact a difference in how liberalized killing policies affect other large carnivores. Further, we hypothesize that it is the attitudes and values of the human actors that are the unifying variable, not the nature of the environmental component or species of wildlife. Our interpretation of these findings, given their consistency with past studies, is that when policies are implemented which reduce the value of non-human beings, such as policies which enable their killing for the sole benefit of human actors, there will be increased harm to those beings and damage to the environment, including crimes.

Data accessibility. The pre-registered Stage 1 report can be found on the Open Science Framework at the following link: https://osf.io/f2kmb/. The raw datasets sources from the US Fish and Wildlife Service and the Office of Law Enforcement have been submitted to the Dryad Digital Repository and can be found at https://doi.org/10.5061/dryad.0vt4b8gxk [71]. We have also included the prepared data in our Dryad Digital Repository submission which is ready to be run through the STATA code included starting on p. 11 of electronic supplementary material.

Authors' contributions. All authors developed the study. The stage 1 manuscript was drafted by F.J.S.-Á. and N.X.L., and was edited and reviewed by all authors. Data analyses were conducted by F.J.S.-Á. and N.X.L., and independent interpretation of the results was first conducted by D.R.P. and A.T. Final interpretation and discussion of results was conducted by all authors. The discussion was drafted by N.X.L. and A.T., with edits and reviews conducted by all authors. All authors have approved the final version of the manuscript.

Competing interests. F.J.S.-Á., N.X.L. and D.R.P. declare no competing interests. A.T. declares no competing interests, and provides his CV (http://faculty.nelson.wisc.edu/treves/archive_BAS/Treves_vita_Jan2020.pdf) and all funding awarded as of 6 January 2020 (http://faculty.nelson.wisc.edu/treves/archive_BAS/funding.pdf) for transparency, so readers can decide if they perceive a competing interest.

Funding. We received funding from the UCLA Law School Animal Law and Policy Grants Program and There Foundation.

Acknowledgements. We thank the US Fish and Wildlife Service Mexican Wolf Recovery Program and Office of Law Enforcement staff, especially Maggie Dwire and John Oakleaf, for data collection, provision and assistance in data interpretation. We thank Judy Calman for assistance in obtaining agency data. We thank the UCLA Law School Animal Law and Policy Grants Program and Therese Foundation for funding. We thank the editor and two anonymous reviewers for their helpful comments and suggestions.

Disclaimer. This article does not necessarily reflect the views of the institutions or agencies involved.

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
