## [Peer Review File · Royal Society Open Science]

Review History

Decision letter (RSOS-200202.R0)

21-Feb-2020

Dear Dr Santiago-Ávila,

I write you in regards to manuscript RSOS-200202 entitled "The effect of allowing lethal management on illegal killing of Mexican wolves" which you submitted to Royal Society Open Science.

We routinely triage submissions for scientific soundness, clarity and general adherence to the Registered Reports guidelines. For submissions that have promise but are not yet suitable for in-depth Stage 1 review, we offer feedback to help authors maximise the chances that reviewers will respond positively to a resubmission.

We have concluded that your submission is not yet suitable for in-depth review and has therefore been rejected at this time, but we believe it will be suitable once several issues are addressed. We

therefore invite a resubmission. Further comments from the Associate Editor may be found at the end of this letter.

If you wish to revise your manuscript in light of the below comments please submit your manuscript as a new submission and mention this previous manuscript ID in your covering letter. You should also provide a detailed response to the below comments in the cover letter.

Please note that Royal Society Open Science will introduce article processing charges for all new submissions received from 1 January 2018. Registered Reports submitted and accepted after this date will ONLY be subject to a charge if they subsequently progress to and are accepted as Stage 2 Registered Reports. If your manuscript is submitted and accepted for publication after 1 January 2018 (i.e. as a full Stage 2 Registered Report), you will be asked to pay the article processing charge, unless you request a waiver and this is approved by Royal Society Publishing. You can find out more about the charges at <https://royalsocietypublishing.org/rsos/charges>. Should you have any queries, please contact openscience@royalsociety.org.

Thank you for considering Royal Society Open Science for the publication of your registered report.

on behalf of Professor Chris Chambers (Registered Reports Editor, Royal Society Open Science)
openscience@royalsociety.org

Associate Editor Comments to Author:

Associate Editor

Comments to the Author:

The proposal is promising but needs some refinement to ensure a direct and precise correspondence between the hypotheses, the critical statistical tests or test components that will test those hypotheses, if appropriate the statistical power of each test or test component, and the interpretation given different outcomes. To ensure maximum clarity please include in the Method section a design table as shown in section 9 of this template (<https://osf.io/93znh/>) outlining the research question, hypothesis, sampling plan, analysis plan, and contingent interpretation.

Author's Response to Decision Letter for (RSOS-200202.R0)

See Appendix A.

RSOS-200330.R0

Review form: Reviewer 1 (Scott Creel)

Do you have any ethical concerns with this paper?

No

Recommendation?

Accept with minor revision

Comments to the Author(s)

See attached PDF (Appendix B). My apologies for the slow review but I have had to flip my classes to online in response to the pandemic.

Review form: Reviewer 2

Do you have any ethical concerns with this paper?

No

Recommendation?

Accept with minor revision

Comments to the Author(s)

As far as I can judge the study is well motivated, and addresses its research question appropriately. I have some queries about the analysis however. A crucial part of the paper is Table 4 which aligns statistical tests with hypotheses. I was unclear about the notation "poa"; it may be obvious, but it should be defined. Also the precise test and criteria should be given in each case in the table so there is no room for analytic flexibility. Finally, the authors should consider how they would know that both hypotheses were false; i.e. the policy had no meaningful effect either way (non-significance does not indicate this; they would need a Bayes factor or equivalence testing etc if they wanted to assert policy change was ineffectual.)

Decision letter (RSOS-200330.R0)

16-Apr-2020

Dear Dr Santiago-Ávila

On behalf of the Editors, I am pleased to inform you that your Manuscript RSOS-200330 entitled "The effect of allowing lethal management on illegal killing of Mexican wolves" has been accepted in principle for publication in Royal Society Open Science subject to minor revision in accordance with the referee and editor suggestions. Please find their comments at the end of this email.

The reviewers and handling editors have recommended publication, but also suggest some minor revisions to your manuscript. Therefore, I invite you to respond to the comments and revise your manuscript.

Please you submit the revised version of your manuscript within 30 days (i.e. by the 24-Apr-2020). If you do not think you will be able to meet this date please let me know immediately.

To revise your manuscript, log into <https://mc.manuscriptcentral.com/rsos> and enter your Author Centre, where you will find your manuscript title listed under "Manuscripts with Decisions". Under "Actions," click on "Create a Revision." You will be unable to make your

revisions on the originally submitted version of the manuscript. Instead, revise your manuscript and upload a new version through your Author Centre.

Full author guidelines can be found here <https://royalsocietypublishing.org/rsos/registered-reports#ReviewerGuideRegRep>.

on behalf of Professor Chris Chambers (Subject Editor, Royal Society Open Science)
openscience@royalsociety.org

Associate Editor Comments to Author (Professor Chris Chambers):

Associate Editor: 1

Comments to the Author:

At the outset, thank you for your patience while we completed the initial review process for your submission -- the coronavirus pandemic has caused significant delays and disruption to editorial and peer review processes across the journal. Two expert reviewers have now appraised the manuscript, including one field expert (Reviewer 1) and one statistical expert (Reviewer 2). Both reviewers are positive about the proposal while also highlighting areas of the manuscript that would benefit from revision. Reviewer 1 offers a very detailed and constructive critique, at a broad level highlighting the risk of potential confounds in the analysis plan and the need for greater methodological detail. Reviewer 2 notes the need for increased precision in the analysis plan and design table (Table 4), specifying precisely which outcomes would or would not support the hypotheses. In relation to Reviewer 2's comment regarding equivalence testing and Bayes factors (and the inherent problem in intending to conclude evidence of no effect statistically non-significant results), the following references may be useful:

<https://www.ncbi.nlm.nih.gov/pmc/articles/PMC5502906/>

<https://www.ncbi.nlm.nih.gov/pmc/articles/PMC4114196/>

Please respond point-by-point to reach concern raised by the reviewers.

Reviewer comments to Author:

Reviewer: 1

Comments to the Author(s)

See attached PDF. My apologies for the slow review but I have had to flip my classes to online in response to the pandemic.

Reviewer: 2

Comments to the Author(s)

As far as I can judge the study is well motivated, and addresses its research question appropriately. I have some queries about the analysis however. A crucial part of the paper is Table 4 which aligns statistical tests with hypotheses. I was unclear about the notation "poa"; it may be obvious, but it should be defined. Also the precise test and criteria should be given in each case in the table so there is no room for analytic flexibility. Finally, the authors should consider how they would know that both hypotheses were false; i.e. the policy had no meaningful effect either way (non-significance does not indicate this; they would need a Bayes factor or equivalence testing etc if they wanted to assert policy change was ineffectual.)

Author's Response to Decision Letter for (RSOS-200330.R0)

See Appendix C.

RSOS-200330.R1

Review form: Reviewer 2

Do you have any ethical concerns with this paper?

No

Recommendation?

Accept with minor revision

Comments to the Author(s)

The authors have addressed my point by using Bayes factors. The model of H1 is specified, and I am happy with that. Be explicit about how the SE of the regression coefficients will be determined, whether the normal approximation will be assumed, and hence also what code/calculator will be used for the Bayes factor (e.g. will a normal likelihood function be assumed). I would advise checking the robustness of the Bayesian tests e.g. with "robustness region" of Dienes (2019).

Dienes, Z. (2019). How do I know what my theory predicts? *Advances in Methods and Practices in Psychological Science*, 2, 364-377. <https://doi.org/10.1177/2515245919876960>

" If results from our endpoints of interest (reported poached) are contradictory or inconclusive, we will proceed to aggregate them and run all analysis363 on the new endpoint of poa+LTF (including BFs). "

Specify exactly what contradictory or inconclusive means, for example what size Bayes factor? Be explicit about what will count against each hypothesis, e.g. what size BF in favour of H0 will be taken not as inconclusive but as good enough evidence against each theory so no need to aggregate. Alternatively, for simplicity, why not just go with the aggregate end point as the basic test? Still be clear about what would count against such a hypothesis, of course.

Decision letter (RSOS-200330.R1)

Dear Dr Santiago-Ávila,

On behalf of the Editors, I am pleased to inform you that your Manuscript RSOS-200330.R1 entitled "Evaluating how lethal management affects illegal killing of Mexican wolves" has been accepted in principle for publication in Royal Society Open Science subject to minor revision in accordance with the referee and editor suggestions. Please find their comments at the end of this email.

The reviewers and handling editors have recommended publication, but also suggest some minor revisions to your manuscript. Therefore, I invite you to respond to the comments and revise your manuscript.

Please you submit the revised version of your manuscript within 7 days (i.e. by the 01-Jul-2020). If you do not think you will be able to meet this date please let me know immediately.

When submitting your revised manuscript, you will be able to respond to the comments made by the referees and you should upload a file "Response to Referees". You can use this to document any changes you make to the original manuscript. In order to expedite the processing of the revised manuscript, please be as specific as possible in your response to the referees.

Full author guidelines can be found here <https://royalsocietypublishing.org/rsos/registered-reports>.

on behalf of Professor Chris Chambers (Subject Editor, Royal Society Open Science)
openscience@royalsociety.org

Associate Editor Comments to Author (Professor Chris Chambers):

The revised manuscript was returned to one of the reviewers, who is largely satisfied but recommends some minor revisions to provide greater detail about the Bayesian analysis plans and ensure falsifiability. These suggestions strike me as very sensible, and provided the authors

are able to respond comprehensively in a final revision, in-principle acceptance should be awarded without requiring further in-depth Stage 1 review.

Reviewer comments to Author:

Reviewer: 2

Comments to the Author(s)

The authors have addressed my point by using Bayes factors. The model of H1 is specified, and I am happy with that. Be explicit about how the SE of the regression coefficients will be determined, whether the normal approximation will be assumed, and hence also what code/calculator will be used for the Bayes factor (e.g. will a normal likelihood function be assumed). I would advise checking the robustness of the Bayesian tests e.g. with "robustness region" of Dienes (2019).

Dienes, Z. (2019). How do I know what my theory predicts? *Advances in Methods and Practices in Psychological Science*, 2, 364-377. <https://doi.org/10.1177/2515245919876960>

" If results from our endpoints of interest (reported poached) are contradictory or inconclusive, we will proceed to aggregate them and run all analysis on the new endpoint of poa+LTF (including BFs). "

Specify exactly what contradictory or inconclusive means, for example what size Bayes factor? Be explicit about what will count against each hypothesis, e.g. what size BF in favour of H0 will be taken not as inconclusive but as good enough evidence against each theory so no need to aggregate. Alternatively, for simplicity, why not just go with the aggregate end point as the basic test? Still be clear about what would count against such a hypothesis, of course.

Author's Response to Decision Letter for (RSOS-200330.R1)

See Appendix D.

Decision letter (RSOS-200330.R2)

Dear Dr Santiago-Ávila

On behalf of the Editor, I am pleased to inform you that your Manuscript RSOS-200330.R2 entitled "Evaluating how lethal management affects poaching of Mexican wolves" has been accepted in principle for publication in Royal Society Open Science.

You may now progress to Stage 2 and complete the study as approved. Before commencing data collection we ask that you:

- 1) Update the journal office as to the anticipated completion date of your study.
- 2) Register your approved protocol on the Open Science Framework (using the dedicated RR registration service at <https://osf.io/rr>), either publicly or privately under embargo until submission of the Stage 2 manuscript. Please note that a time-stamped, independent registration

of the protocol is mandatory under journal policy, and manuscripts that do not conform to this requirement cannot be considered at Stage 2. The protocol should be registered unchanged from its current approved state, with the time-stamp preceding implementation of the approved study design. Please include a URL to the protocol in your Stage 2 manuscript (e.g. 'This article received in-principle acceptance (IPA) at Royal Society Open Science on 6 July, 2020. Following IPA, the accepted Stage 1 version of the manuscript was preregistered on the OSF (URL). This preregistration was performed prior to data analysis.')

Following completion of your study, we invite you to resubmit your paper for peer review as a Stage 2 Registered Report. Please note that your manuscript can still be rejected for publication at Stage 2 if the Editors consider any of the following conditions to be met:

- The results were unable to test the authors' proposed hypotheses by failing to meet the approved outcome-neutral criteria.
- The authors altered the Introduction, rationale, or hypotheses, as approved in the Stage 1 submission.
- The authors failed to adhere closely to the registered experimental procedures. Please note that any deviations from the approved experimental procedures must be communicated to the editor immediately for approval, and prior to the completion of data collection. Failure to do so can result in revocation of in-principle acceptance and rejection at Stage 2 (see complete guidelines for further information).
- Any post-hoc (unregistered) analyses were either unjustified, insufficiently caveated, or overly dominant in shaping the authors' conclusions.
- The authors' conclusions were not justified given the data obtained.

We encourage you to read the complete guidelines for authors concerning Stage 2 submissions at <https://royalsocietypublishing.org/rsos/registered-reports#ReviewerGuideRegRep>. Please especially note the requirements for data sharing, reporting the URL of the independently registered protocol, and that withdrawing your manuscript will result in publication of a Withdrawn Registration.

Please note that Royal Society Open Science will introduce article processing charges for all new submissions received from 1 January 2018. Registered Reports submitted and accepted after this date will ONLY be subject to a charge if they subsequently progress to and are accepted as Stage 2 Registered Reports. If your manuscript is submitted and accepted for publication after 1 January 2018 (i.e. as a full Stage 2 Registered Report), you will be asked to pay the article processing charge, unless you request a waiver and this is approved by Royal Society Publishing. You can find out more about the charges at <https://royalsocietypublishing.org/rsos/charges>. Should you have any queries, please contact openscience@royalsociety.org.

Once again, thank you for submitting your manuscript to Royal Society Open Science and we look forward to receiving your Stage 2 submission. If you have any questions at all, please do not hesitate to get in touch. We look forward to hearing from you shortly with the anticipated submission date for your stage two manuscript.

on behalf of Professor Chris Chambers (Registered Reports Editor, Royal Society Open Science)
openscience@royalsociety.org

Author's Response to Decision Letter for (RSOS-200330.R2)

See Appendix E.

RSOS-200330.R3 (Revision)

Review form: Reviewer 2

Is the manuscript scientifically sound in its present form?

No

Are the interpretations and conclusions justified by the results?

No

Is the language acceptable?

Yes

Do you have any ethical concerns with this paper?

No

Have you any concerns about statistical analyses in this paper?

Yes

Recommendation?

Accept with minor revision

Comments to the Author(s)

The manuscript is clear and well written. However, the authors need to strictly adhere to the logic of a registered report. Specifically the analyses need a section where the planned analyses are given, whether they pan out as appropriate or not; and then a different clearly labelled section "non-planned" or "exploratory", even if the non-planned analyses were well justified. "Non-planned" does not mean that the analyses are suspect; just that the reader needs to know they were not planned, including readers who just read e.g. abstracts or discussions.

P 25: "Covariates of winter and sex did not significantly affect the results of any models, and therefore the most parsimonious model included the policy intervention without either covariate."

I could not find a statement in the planned analyses that non-significant co-variates would be removed. Thus, for the planned analyses I see no justification for removing these covariates. (They could be reported in a non-planned analyses section).

Planned analyses referred to inference by Bayes Factors, specifically deciding according to whether 2 out of 3 pointed in a certain direction. The planned analysis section should stick to this without using p-values for inference. The DV should the sum of POA+LTF. Then an exploratory analysis section can be introduced, that digs into different DVs more, as the authors have done, changes the use of BFs or uses p-values.

In the discussion first consider the planned analyses then make explicit what follows from non-planned analyses.

Do not assert no effect (lines 575, 613) without evidence for it (e.g. a relevant Bayes factor, even if unplanned)

The abstract should first report conclusions from planned analyses and then indicate any other conclusions it reports follow from "exploratory" or "unplanned" analyses.

Decision letter (RSOS-200330.R3)

Dear Dr Santiago-Ávila:

On behalf of the Editor, I am pleased to inform you that your Stage 2 Registered Report RSOS-200330.R3 entitled "Evaluating how lethal management affects poaching of Mexican wolves" has been deemed suitable for publication in Royal Society Open Science subject to minor revision in accordance with the referee suggestions. Please find the referees' comments at the end of this email.

The reviewers and Subject Editor have recommended publication, but also suggest some minor revisions to your manuscript. Therefore, I invite you to respond to the comments and revise your manuscript.

Please also ensure that all the below editorial sections are included where appropriate -- if any section is not applicable to your manuscript, please can we ask you to nevertheless include the heading, but explicitly state that the heading is inapplicable. An example of these sections is attached with this email.

- Ethics statement

- Data accessibility

If you wish to submit your supporting data or code to Dryad (<http://datadryad.org/>), or modify your current submission to dryad, please use the following link:
[http://datadryad.org/submit?journalID=RSOS&manu=\(Document not available\)](http://datadryad.org/submit?journalID=RSOS&manu=(Document not available))

- Competing interests

- Authors' contributions

- Acknowledgements

- Funding statement

Because the schedule for publication is very tight, it is a condition of publication that you submit the revised version of your manuscript within 21 days (i.e. by the 11-Feb-2021). If you do not think you will be able to meet this date please let me know immediately.

- 1) A text file of the manuscript (tex, txt, rtf, docx or doc), references, tables (including captions) and figure captions. Do not upload a PDF as your "Main Document".
- 2) A separate electronic file of each figure (EPS or print-quality PDF preferred (either format should be produced directly from original creation package), or original software format)
- 3) Included a 100 word media summary of your paper when requested at submission. Please ensure you have entered correct contact details (email, institution and telephone) in your user account
- 4) Included the raw data to support the claims made in your paper. You can either include your data as electronic supplementary material or upload to a repository and include the relevant doi within your manuscript

5) All supplementary materials accompanying an accepted article will be treated as in their final form. Note that the Royal Society will neither edit nor typeset supplementary material and it will be hosted as provided. Please ensure that the supplementary material includes the paper details where possible (authors, article title, journal name).

on behalf of Professor Chris Chambers
(Registered Reports Editor, Royal Society Open Science)
openscience@royalsociety.org

Associate Editor Comments to Author (Professor Chris Chambers):

Associate Editor: 1

Comments to the Author:

One of the original Stage 1 reviewers was available to review the Stage 2 manuscript. To accelerate the Stage 2 review process, I am issuing an interim Revise decision now to address the core points addressed in this review, which are fundamental to the Registered Reports format. Provided these issues are thoroughly addressed, the revised manuscript will then be considered by field specialist editors in a final assessment without sending to additional reviewers.

In revising, please pay special attention to the requirement to carefully distinguish the outcomes of preregistered analyses from unplanned (post hoc) analyses; this includes any analysis decisions that were not prespecified at Stage 1. Please also ensure that this level of reporting transparency is also applied to the Abstract.

Comments to Author:

Reviewer: 2

Comments to the Author(s)

The manuscript is clear and well written. However, the authors need to strictly adhere to the logic of a registered report. Specifically the analyses need a section where the planned analyses are given, whether they pan out as appropriate or not; and then a different clearly labelled section "non-planned" or "exploratory", even if the non-planned analyses were well justified. "Non-planned" does not mean that the analyses are suspect; just that the reader needs to know they were not planned, including readers who just read e.g. abstracts or discussions.

P 25: "Covariates of winter and sex did not significantly affect the results of any models, and therefore the most parsimonious model included the policy intervention without either covariate."

I could not find a statement in the planned analyses that non-significant co-variates would be removed. Thus, for the planned analyses I see no justification for removing these covariates. (They could be reported in a non-planned analyses section).

Planned analyses referred to inference by Bayes Factors, specifically deciding according to whether 2 out of 3 pointed in a certain direction. The planned analysis section should stick to this without using p-values for inference. The DV should be the sum of POA+LTF. Then an exploratory analysis section can be introduced, that digs into different DVs more, as the authors have done, changes the use of BFs or uses p-values.

In the discussion first consider the planned analyses then make explicit what follows from non-planned analyses.

Do not assert no effect (lines 575, 613) without evidence for it (e.g. a relevant Bayes factor, even if unplanned)

The abstract should first report conclusions from planned analyses and then indicate any other conclusions it reports follow from "exploratory" or "unplanned" analyses.

Author's Response to Decision Letter for (RSOS-200330.R3)

See Appendix F.

Decision letter (RSOS-200330.R4)

Dear Dr Santiago-Ávila:

It is a pleasure to accept your revised Stage 2 Registered Report entitled "Evaluating how lethal management affects poaching of Mexican wolves" in its current form for publication in Royal Society Open Science.

There are two final (minor) changes to make at the copyediting state:

1. The section "Timeline" can be removed. Specifically, the text below is not essential to include in the Stage 2 published article:

"Timeline

We are able to start work immediately following acceptance. Our anticipated timeline for completion pending acceptance of our submission is the following: 4-5 weeks to conduct all statistical analyses and 3-4 weeks for writing and submission."

2. Please now make all OSF project components public and replace any private view-only OSF URLs with their public URLs.

These changes can be made at the proof stage and need not require a formal resubmission.

You can expect to receive a proof of your article in the near future. Please contact the editorial office (openscience@royalsociety.org) and the production

office (openscience_proofs@royalsociety.org) to let us know if you are likely to be away from e-mail contact – if you are going to be away, please nominate a co-author (if available) to manage the proofing process, and ensure they are copied into your email to the journal.

on behalf of Professor Chris Chambers (Subject Editor)
openscience@royalsociety.org

Appendix A

Response to Decision letter

We are grateful for the opportunity to revise and resubmit and have done our best to address the associate editor's advice. We have also added the below responses to our Cover Letter, as requested in the decision letter.

As per the associated editor's advice, we have refined the proposal with special attention to clarifying the relationship between our hypotheses, proposed analyses and interpretation of outcomes (including contingent interpretation and synthesis of model results). This has all been addressed in the text and with the addition of a new Table (#4) modelled after that in Section 9 of the OSF template. We have also addressed a statement to certify our models comply with the appropriate number of observations for obtaining precise estimates and associated quantities. We hope this clarifies our proposal and complies with your comments. Below, we also note where the answers to each of the questions in the OSF template can be found in our proposal:

Q1 – Our main question can be found in Background and Table 4.

Q2 – Endpoints and main independent variable are identified and described in the Methods (Data collection and preparation section) and Table 4.

Q3 – The competing hypotheses we are testing for, as described in the literature, can be found in the Background section ('killing for tolerance' and 'facilitated illegal killing'). We have drawn a direct correspondence between them and our analyses in Table 4.

Q4 – The data we will use has already been collected by government agencies.

Q5 – A description of the data already in hand and rules used for determining study period(s) and data exclusion criteria for our analyses can be found in the Methods (Data collection and preparation section) and Table 4. We have also identified and described our pending data request for extending our study period within the same section, and have described the steps we will take to include these data and how they would strengthen our inferences.

Q6 - A description of rules used for inclusion of observations in our analyses can be found in the Methods (Data collection and preparation section) and Table 4.

Q7 – See Q5 above.

Q8 – See Methods (Analysis section)

Q9 – See Methods and Table 4.

Q10 – Throughout the proposal, we are clear on intending to analyze existing data provided by US federal agencies.

Thank you for considering and with kind regards,

Francisco J. Santiago-Ávila, Naomi Louchouart & Adrian Treves

Carnivore Coexistence Lab: <http://faculty.nelson.wisc.edu/treves/>
Nelson Institute for Environmental Studies, University of Wisconsin-Madison
santiagoavil@wisc.edu • Tel. (919) 530-9280

Appendix B

Summary Comments:

1. The scientific validity of the research question(s).

The study addresses an important and relatively little-studied question, whether policies that promote legal killing of wolves reduce the likelihood of illegal killing.

2. The logic, rationale, and plausibility of the proposed hypotheses.

The authors consider a pair of logical but mutually exclusive hypotheses, that an increase legal killing increases or decreases the rate of illegal killing, and use the existing literature to lay out the underlying rationale for each.

3. The soundness and feasibility of the methodology and analysis pipeline (including statistical power analysis where applicable).

The analytic methods are sound and feasible. See detailed comments below for concerns about:

- A. the possibility that other environmental causes of death might temporally covary with changes in policy and lead to mistaken inference about the effects of policy on mortality, and
- B. a limited number of deaths might lead to strong signatures in the fitted hazard functions from a small number of events
- C. the deaths tabulated in Tables 2 & 3 may not all represent independent events

4. Whether the clarity and degree of methodological detail would be sufficient to replicate the proposed experimental procedures and analysis pipeline.

With regard to analytic methods, the clarity is good and would allow replicated analysis. With regard to the original field methods used to obtain the monitoring data, description is almost completely absent and could not be replicated. For example I did not see mention of whether individuals were monitored with VHF, GPS, satellite GPS collars or a combination of those. Such methodological details could have large effects on the results.

5. Whether the authors provide a sufficiently clear and detailed description of the methods to prevent undisclosed flexibility in the experimental procedures or analysis pipeline.

The description is sufficiently clear and detailed.

6. Whether the authors have considered sufficient outcome-neutral conditions (e.g. absence of floor or ceiling effects; positive controls; other quality checks) for ensuring that the results obtained are able to test the stated hypotheses.

See response 3A and detailed comments below. There remains some concern that any environmental effect on population size or mortality that coincided with the middle of three time periods with respect to policy could be mistaken for an effect of policy.

Detailed comments:

Line 7-8, Lines 28-31: While this comment pertains only to the broad framing of the study, such as the passages noted here, it is important to distinguish between illegal killing as an immediate cause of death and more diffuse problems that are much harder to assign as a cause of death (habitat degradation and loss, prey depletion, human encroachment on protected areas). For example, there is simply no immediate mortality to be observed in places where carnivores have been fully extirpated by habitat conversion. It is also perhaps important to acknowledge at the outset that these limiting factors are probably not independent in their effects on carnivore demography and dynamics.

Line 15: It would be useful to state more explicitly what is meant by 'data on monitoring'. This could mean many different things: rigorous estimates of population size, an index of population size, survival from individual resighting data.

Line 35: These prior efforts to quantify illegal killing and effects on it are important, but I believe this approach is vulnerable to a general problem that arises when changes through time are attributed to a specific causal mechanism: any variable that is temporally correlated with the causal variable will appear causal in a univariate analysis. Ways to strengthen inference could include collecting data over periods with multiple changes in policy (as in the Chapron & Treves paper discussed at line 69), statistically controlling other variables that may affect survival, and comparing temporal changes in survival rates for otherwise-matched populations that did/did not experience a change in policy (for a pseudo-BACI design). In concept, all of these approaches can be combined. For this study, statistical control of other variables that might temporally covary with policy changes is probably the most relevant. If policy changes only once, then any variable with a time trend will tend to covary with it.

Lines 43-44: "...did not directly estimate illegal killing, instead using scalars from other populations..." It would be useful to more precisely state what is meant by 'scalars from other populations'.

Lines 55- 59: "We call this first hypothesis 'killing for tolerance', which predicts legal killing will reduce illegal killing through the following mechanism: legalizing or liberalizing killing of controversial species will lead would-be perpetrators to desist from illegal killing because of increased tolerance for the species or protectionist policies." Perhaps this could be clarified by noting explicitly that reducing population size by legal killing might reduce illegal killing rates (e.g. results cited at line 85 onward) whether it alters 'tolerance' or not. To clarify this issue, it would be useful to define tolerance explicitly at this point. Is tolerance equivalent to a reduction in illegal killing, or does it imply something broader? Line 73 states that changes in tolerance can be measured by social surveys, which implies that tolerance is not being equated directly to reduced illegal killing.

Lines 72-78: this sentence is important for the logic of the paper, but long and difficult to follow.

Line 104: See comment on line 35 with respect to statistical control of other variables that might temporally covary with policy changes.

Line 113-114: grammar

Lines 137-141: It would be useful to state whether these are VHF, GPS or satellite GPS collars. If either of the latter two, provide the fix rate. If the former, provide some information on the monitoring methods, effort and realized distribution of inter-fix intervals. It would also be useful to know the model

of radiocollar, to allow some intuition about the likelihood of collar failure prior to expected battery failure as a reason for the 'lost to follow up' (LTF) endpoint. Some data on the fraction of LTF endpoints that fell within some defined window prior to the expected end of battery life would be of value in this regard. Data on the fraction of collars that drifted in frequency or (if GPS) dropped in fix rate (or any other indicator of problems) prior to an LTF endpoint would also be of value.

Line 168 – 169: '...censoring LTF led to systematic under-estimates of the rate and hazard of illegal killing..' I think the focal advance intended from this work is to provide refined estimates of this issue, and I think this statement of the point is clearer than the prior discussion. Perhaps it would be useful to state the issue in this way earlier in the development of the question/approach.

Line 205: See prior comments on other variables that might affect wolf survival but temporally covary with policy. Obtaining existing data on one more change of policy would greatly strengthen inference, as would statistical control of environmental covariates (beyond season) that might temporally covary with a small number (2) of policy changes.

Figure 1: Explain why large, distinct shifts in cumulative hazard occur on the same day of monitoring for both policies (high and low legal killing). It seems (to me) unlikely that cumulative hazard really does double exactly on day 2400 (or whatever that day is – hard to identify precisely from the figure) under both policies. It seems more likely that this is an artefact of the data or model structure. I suspect that this is a product of fitting the hazard model to 32 mortalities, so that a single event with more than one death is having a pronounced effect? This raises a broader issue: even with a large sample of live wolves, the number of mortalities may be small enough (Table 2) that a small number of events carry a strong signature in the results. Figure 1 present data from 513 wolves, and the sample for this study is 168 wolves, suggesting that this issue will be more important here.

Table 2: Are the mortalities tabulated here independent events, or are their instances in which multiple packmates died on the same day (especially for rows 1,3 and 4)? Same comment on Table 3, and see comment just above.

Figure 2. Explain why the pronounced step in the CHF of Figure 1 (see last comment) becomes undetectable in the CIFs fit to the same data in Figure 2?

Line 322: grammar

Appendix C

Response to Reviewers

Associate Editor Comments to Author (Professor Chris Chambers):

Associate Editor: 1

Comments to the Author:

At the outset, thank you for your patience while we completed the initial review process for your submission -- the coronavirus pandemic has caused significant delays and disruption to editorial and peer review processes across the journal. Two expert reviewers have now appraised the manuscript, including one field expert (Reviewer 1) and one statistical expert (Reviewer 2). Both reviewers are positive about the proposal while also highlighting areas of the manuscript that would benefit from revision. Reviewer 1 offers a very detailed and constructive critique, at a broad level highlighting the risk of potential confounds in the analysis plan and the need for greater methodological detail. Reviewer 2 notes the need for increased precision in the analysis plan and design table (Table 4), specifying precisely which outcomes would or would not support the hypotheses. In relation to Reviewer 2's comment regarding equivalence testing and Bayes factors (and the inherent problem in intending to conclude evidence of no effect statistically non-significant results), the following references may be useful:

<https://www.ncbi.nlm.nih.gov/pmc/articles/PMC5502906/>

<https://www.ncbi.nlm.nih.gov/pmc/articles/PMC4114196/>

Please respond point-by-point to reach concern raised by the reviewers.

Response: Thank you for the opportunity to revise our proposal considering the reviewer comments. We are grateful for the editor and reviewer's insightful and constructive critique as well as their recommendations for addressing any limitations.

Reviewer comments to Author:

Reviewer: 1

Summary Comments:

1. The scientific validity of the research question(s).

The study addresses an important and relatively little-studied question, whether policies that promote legal killing of wolves reduce the likelihood of illegal killing.

2. The logic, rationale, and plausibility of the proposed hypotheses.

The authors consider a pair of logical but mutually exclusive hypotheses, that an increase legal killing increases or decreases the rate of illegal killing, and use the existing literature to lay out the underlying rationale for each.

3. The soundness and feasibility of the methodology and analysis pipeline (including statistical power analysis where applicable).

The analytic methods are sound and feasible. See detailed comments below for concerns about:

A. the possibility that other environmental causes of death might temporally covary with changes in policy and lead to mistaken inference about the effects of policy on mortality, and

Response: Although we cannot rule out a nuisance variable coinciding with the policy on and off periods, our analysis is more robust than a before-during-after comparison of a single replicate. The nuisance variable would have to be widespread across both NM and AZ, multiple jurisdictions (tribal, state, federal, county lands), and affecting multiple independent adult wolves in packs occupying virtually exclusive home ranges. That leaves a climatic event or other widespread biotic event such as a disease. We have no evidence for such events but we re-examined rainfall records and while there were droughts throughout the study period, there was no evidence of a persistent drought that specifically affected the policy period or coincided with its start or end (US Drought Monitor). Moreover, we have searched the annual reports of the Mexican Wolf Recovery Program and found no mention of changes in environmental events or onsets of disease. Moreover, our analysis is more robust because we have 88, 81 and 52 replicates within each policy period, and 46 wolves (up to 2012) that survived through transitions in policy (one transition $n=39$ and two transitions $n=7$). Therefore, we have many replicated life histories and two replicated changes in policy for those individual wolves.

Furthermore, any nuisance event that occurred would need to affect the poaching, LTF and legal killing risks in order to affect our hypotheses. A natural event such as a long lasting climatic event is unlikely to have done so, instead affecting our 'natural' endpoint. Indeed, environmental changes that may covary with the policy may in fact show changes to the 'natural' endpoint hazard and incidence, while perhaps affecting the changes in incidence of our anthropogenic endpoints but not their hazards. We have briefly added the above discussion to our methods section.

B. a limited number of deaths might lead to strong signatures in the fitted hazard functions from a small number of events

We agree with the concern. That said, we are following best practices of having at least 10 endpoint events per covariate for our current dataset up to 2012 (Concato et al. 1995; Peduzzi et al. 1995; Vittinghoff and McCulloch 2007). Admittedly, this may be an issue for the 'natural' endpoint, but this is not relevant to our hypotheses. The limit on number of events also relates to the addition of other (nuisance) covariates (above). We would not like to further decrease our number of events per covariate without significant evidence that the omission will bias our results. So far, we have not found any additional covariates that we believe fit these criteria. We expect to further mitigate this issue by including data up to 2016, which will include 4 more years of wolf records.

C. the deaths tabulated in Tables 2 & 3 may not all represent independent events
If much of our sample had been collared or re-collared near in time this would confound our interpretations of policy periods. Or if many of the wolves died or disappeared on the same

day (especially if they were collared on the same day) their survival statistics would be non-independent. We examined the data thanks to the reviewer asking us this important question.

For the tests of our hypotheses, only LTF and poaching are relevant in relation to legal killing (not natural or collisions). Indeed only legal killing is expected to be clustered in time in those rare cases where multiple members of a pack are targeted for removal in the same operation and federal agents can move swiftly and openly from one wolf to the next within a pack. Furthermore, clustering of legal lethal control does not affect the test of our hypotheses, which hinges on the risk and hazard of observed poaching or of LTF. Poachers would not move so openly or swiftly across large areas to our knowledge. Given poaching was a federal offense, we presume poachers would act more discreetly. Therefore, if poaching or LTF were clustered in time it would require a fatal event affecting multiple pack members nearly simultaneously such as poison rather than shooting or trapping. We have found no evidence of use of poison within Mexican wolf records or OLE data. The following summary descriptions show how rarely two wolves reached an endpoint on the same day.

The data reveal that on 51 occasions during the study period, 2 or more wolves were collared on the same day. Packs were always collared on the same day upon their first release from captivity into the wild, therefore collaring of 2 or more wolves on the same day occurred when packs of between 2 and 4 (average 3) wolves were released together. On 13 occasions in the study period, 2 or more wolves reached the same endpoint on the same day. Nine of those 13 occasions were a result of agency removals. Of those 9, only 3 occasions were wolves collared on the same day and then reached the same endpoint on the same day and these were removed legally. Of the 13 occasions in which 2 or more wolves reached the same endpoint on the same day, only two instances were human-caused other than legal and two more were a result of two or more wolves going missing on the same day. So 4 of 279 (1%) wolves in our sample reached an endpoint on the same day because of poaching or LTF and of wolves collared on the same day that died on the same day (5%) all of those were legal removals.

Given that adult wolves often travel alone – pack hunting is reserved for large prey or territorial patrols (Mech 1999, 1970)– the putative observation of multiple deaths or disappearance in a single period would likely reflect human intention rather than natural causes coinciding for multiple independent adult wolves in a pack of 2-4 individuals (after all these are not pups that might get caught together in a den or rendezvous site— USFWS). Therefore, the putative existence of coinciding incidents of observed poaching or LTF actually lends weight to either hypothesis depending on which fate changes and in which direction.

In sum, our results about LTF would be undermined if many collars were affixed nearly simultaneously and they had similar battery lives so wolves went off the air in waves. See above for the rarity of simultaneous endpoints. Therefore, we can conclude that collaring date and collar battery life cannot be the primary cause of LTF, and also that all wolf histories can be considered independent (to the extent possible within the social structure of wolves), making their endpoints also independent.

4. Whether the clarity and degree of methodological detail would be sufficient to replicate the proposed experimental procedures and analysis pipeline.

With regard to analytic methods, the clarity is good and would allow replicated analysis. With regard to the original field methods used to obtain the monitoring data, description is almost completely absent and could not be replicated. For example, I did not see mention of whether individuals were monitored with VHF, GPS, satellite GPS collars or a combination of those. Such methodological details could have large effects on the results.

Wolves were collared with a mixture of VHF, GPS and Satellite GPS collars. However, the vast majority of spells (intervals of monitoring history) were obtained via VHF collars (87.6% of spells).

We have added the above statement starting on line 184.

5. Whether the authors provide a sufficiently clear and detailed description of the methods to prevent undisclosed flexibility in the experimental procedures or analysis pipeline.

The description is sufficiently clear and detailed.

6. Whether the authors have considered sufficient outcome-neutral conditions (e.g. absence of floor or ceiling effects; positive controls; other quality checks) for ensuring that the results obtained are able to test the stated hypotheses.

See response 3A and detailed comments below. There remains some concern that any environmental effect on population size or mortality that coincided with the middle of three time periods with respect to policy could be mistaken for an effect of policy.

See response to comment 3A.

Detailed comments:

Line 7-8, Lines 28-31: While this comment pertains only to the broad framing of the study, such as the passages noted here, it is important to distinguish between illegal killing as an immediate cause of death and more diffuse problems that are much harder to assign as a cause of death (habitat degradation and loss, prey depletion, human encroachment on protected areas). For example, there is simply no immediate mortality to be observed in places where carnivores have been fully extirpated by habitat conversion. It is also perhaps important to acknowledge at the outset that these limiting factors are probably not independent in their effects on carnivore demography and dynamics.

Thank you for the suggestion. We have included a statement to this effect on lines 28-31 of the manuscript.

Line 15: It would be useful to state more explicitly what is meant by 'data on monitoring'. This could mean many different things: rigorous estimates of population size, an index of population size, survival from individual resighting data.

We agree and have added a statement to that effect on line 15 of the manuscript

Line 35: These prior efforts to quantify illegal killing and effects on it are important, but I believe this approach is vulnerable to a general problem that arises when changes through time are attributed to a specific causal mechanism: any variable that is temporally correlated with the causal variable will appear causal in a univariate analysis. Ways to strengthen inference could include collecting data over periods with multiple changes in policy (as in the Chapron & Treves paper discussed at line 69), statistically controlling other variables that may affect survival, and comparing temporal changes in survival rates for otherwise-matched populations that did/did not experience a change in policy (for a pseudo-BACI design). In concept, all of these approaches can be combined. For this study, statistical control of other variables that might temporally covary with policy changes is probably the most relevant. If policy changes only once, then any variable with a time trend will tend to covary with it.

We are not as concerned about the suggested potential confounding effect for a few reasons. The policy changed twice in this analysis (and 3 times up to 2016), so the search for a nuisance variable is narrowed as explained above and identifying a real confound must show the on again off again pattern even if it does not coincide exactly, not simply change over time. Moreover, the causal mechanism would need to specifically affect the hazard and incidence of our anthropogenic endpoints, rather than the 'natural' one.

Lines 43-44: "...did not directly estimate illegal killing, instead using scalars from other populations..." It would be useful to more precisely state what is meant by 'scalars from other populations'.

We have edited to clarify in line 47.

Lines 55- 59: "We call this first hypothesis 'killing for tolerance', which predicts legal killing will reduce illegal killing through the following mechanism: legalizing or liberalizing killing of controversial species will lead would-be perpetrators to desist from illegal killing because of increased tolerance for the species or protectionist policies." Perhaps this could be clarified by noting explicitly that reducing population size by legal killing might reduce illegal killing rates (e.g. results cited at line 85 onward) whether it alters 'tolerance' or not. To clarify this issue, it would be useful to define tolerance explicitly at this point. Is tolerance equivalent to a reduction in illegal killing, or does it imply something broader?

Thank you for this insightful comment and suggestion. We have added statements in the text on lines 118-127. Although we cannot make direct links to the cognitive mechanisms involved, we are proposing to evaluate the symptoms of tolerance through the exploration of overall changes in anthropogenic mortality and disappearances; that is, mortality components affected by human behavior. Then following up this analysis with a finer scale evaluation to explore through which specific anthropogenic endpoints may be affected.

Line 73 states that changes in tolerance can be measured by social surveys, which implies that tolerance is not being equated directly to reduced illegal killing.

Please see the above comment, which addresses this concern.

Lines 72-78: this sentence is important for the logic of the paper, but long and difficult to follow.

Thank you for pointing this out, we have edited to make it clearer.

Line 104: See comment on line 35 with respect to statistical control of other variables that might temporally covary with policy changes.

We have responded to this concern and suggestions above (comment 3A).

Line 113-114: grammar

Edited.

Lines 137-141: It would be useful to state whether these are VHF, GPS or satellite GPS collars. If either of the latter two, provide the fix rate. If the former, provide some information on the monitoring methods, effort and realized distribution of inter-fix intervals. It would also be useful to know the model of radiocollar, to allow some intuition about the likelihood of collar failure prior to expected battery failure as a reason for the 'lost to follow up' (LTF) endpoint. Some data on the fraction of LTF endpoints that fell within some defined window prior to the expected end of battery life would be of value in this regard. Data on the fraction of collars that drifted in frequency or (if GPS) dropped in fix rate (or any other indicator of problems) prior to an LTF endpoint would also be of value.

We have added a statement in the text starting on line 180 to explain the different collar types, as well as the average time to LTF, range and variability, and expected battery life for each collar type used on our wolf sample. We hope this will suffice, given we had no information beyond what was provided in the data (collar type and recollaring date though not cause) and what we could glean from annual reports. Moreover, these types of mechanical and battery issues with collars resulting in LTFs should affect wolves at more or less the same rate throughout the period given most of the monitoring happened via VHF, without the policy affecting either.

Line 168 – 169: '...censoring LTF led to systematic under-estimates of the rate and hazard of illegal killing.' I think the focal advance intended from this work is to provide refined estimates of this issue, and I think this statement of the point is clearer than the prior discussion. Perhaps it would be useful to state the issue in this way earlier in the development of the question/approach.

Thank you for this suggestion, we have added a statement of the issue on line 156.

Line 205: See prior comments on other variables that might affect wolf survival but temporally covary with policy. Obtaining existing data on one more change of policy would greatly strengthen inference, as would statistical control of environmental covariates (beyond season) that might temporally covary with a small number (2) of policy changes.

See above responses regarding nuisance variables (comment 3A).

Figure 1: Explain why large, distinct shifts in cumulative hazard occur on the same day of monitoring for both policies (high and low legal killing). It seems (to me) unlikely that cumulative hazard really does double exactly on day 2400 (or whatever that day is – hard to identify precisely from the figure) under both policies. It seems more likely that this is an artefact of the data or model structure. I suspect that this is a product of fitting the hazard model to 32 mortalities, so that a single event with more than one death is having a pronounced effect? This raises a broader issue: even with a large sample of live wolves, the number of mortalities may be small enough (Table 2) that a small number of events carry a strong signature in the results. Figure 1 present data from 513 wolves, and the sample for this study is 168 wolves, suggesting that this issue will be more important here.

Thank you for this observation. We have edited the figure caption to clarify. As you stated above, in Fig. 1 the data is fitted only to the subjects with legal mortalities (32 legal killings, not all 513 subjects from WI). However, the reason for this is not due to multiple mortalities occurring on the same day (or event). Rather, both cumulative hazard curves kink on the same days because both illustrate all 32 legal wolf mortalities included in the Cox model. Cox model results allow for the estimation of a non-parametric baseline hazard and a predicted hazard for all subjects, and the flexibility to identify which factor level should be considered as the baseline hazard. In Fig 1., the cumulative *baseline* hazard is the blue line and illustrates the cumulative hazard those wolves face under strict protection. The cumulative *predicted* hazard is illustrated by the red line and represents the predicted hazard of those legally killed wolves under liberalized killing, considering the effect of the policy change as measured by the Cox model hazard ratio (the changes between curves are reflecting the policy HR=3.30 obtained by the Cox model). To this we add that cumulative hazard curves are notoriously hard to interpret given they don't translate directly to probabilities or incidence, and less so in a competing risk framework; they are usually used to illustrate the general shape of the curve (e.g., hazard of being legally killed increasing non-linearly over time for collared wolves) and the relative change based on the HR.

Table 2: Are the mortalities tabulated here independent events, or are their instances in which multiple packmates died on the same day (especially for rows 1,3 and 4)? Same comment on Table 3, and see comment just above.

Please see our previous response to this concern (comment 3C above).

Figure 2. Explain why the pronounced step in the CHF of Figure 1 (see last comment) becomes undetectable in the CIFs fit to the same data in Figure 2?

The pronounced step is not quite undetectable; there is a similar ‘kink’ at precisely the same time (t=2400) and the magnitude of this is similar but not exact. This is because CIFs do not only consider the hazard (not the cumulative hazard) of their particular endpoint; rather, they are a function of all competing risk hazards (because all hazards interact to shape incidence of all endpoints), which is why hazards aren’t directly translatable into incidences. For the ‘legal killing’ endpoint in Fig 2., the FG model estimated an SHR=3.7 increase in that endpoint from liberalizing killing (black dotted line) relative to strict protections (black solid line), so the relative effect (between curves) is more pronounced. It looks less pronounced for points within the same curve because FG models consider all subjects including those experiencing other competing risks (not only those wolves legally killed), thus having the effect of smoothing the curve.

Line 322: grammar

Edited accordingly.

Reviewer: 2

Comments to the Author(s)

As far as I can judge the study is well motivated, and addresses its research question appropriately. I have some queries about the analysis however. A crucial part of the paper is Table 4 which aligns statistical tests with hypotheses. I was unclear about the notation "poa"; it may be obvious, but it should be defined. Also the precise test and criteria should be given in each case in the table so there is no room for analytic flexibility. Finally, the authors should consider how they would know that both hypotheses were false; i.e. the policy had no meaningful effect either way (non-significance does not indicate this; they would need a Bayes factor or equivalence testing etc if they wanted to assert policy change was ineffectual.)

We have clarified the meaning of “poa” in the Table 4. Figure caption.

The reviewer’s request to pre-select a threshold effect size (in either direction) constrains us to a frequentist criterion for significance. We prefer to relate the CIF of LTF and the CIF of poaching to the CIF of legal killing because legal killing is known, perfectly measured hazard intentionally caused by the policy. If LTF or poaching CIF curves are similar in magnitude to the CIF of legal killing then we can be confident there was a side-effect of the policy as predicted by the hypotheses. If the CIFs of LTF and poaching are lower in magnitude than legal killing we can quantify how much lower and discuss them in light of that. Having said this, we agree with the reviewer that minimal changes in LTF or poaching CIFs should not be treated uncritically as support for one or the other hypothesis. We propose instead to examine the CIF of legal killing and its variability as our guide to the magnitude of change one would consider informative for policy-makers. Additionally, we’d like to thank the reviewer for suggesting the use of Bayes factor (BF) as an alternative to assess strength of evidence for

our alternatives and null. We have incorporated the use of BF following Dienes 2014 (kindly recommended by Prof. Chambers) and based on the estimates obtained from our legal endpoint as the only perfectly reported category (i.e., the higher reliability of its point estimate).

References

Concato, John, Peter Peduzzi, Theodore R. Holford, and Alvan R Feinstein. 1995. "Importance of Events per Independent Variable in Proportional Hazards Analysis I. Background, Goals and General Strategy." *Journal of Clinical Epidemiology* 48 (12): 1495–1501.

Mech, David L. 1970. "The Wolf: The Ecology and Behavior of an Endangered Species. The Nat. Must. Press. Garden City."

———. 1999. "Alpha Status, Dominance, and Division of Labor in Wolf Packs." *Canadian Journal of Zoology* 77 (8): 1196–1203.

Peduzzi, Peter, John Concato, Alvan R Feinstein, and Theodore R. Holford. 1995. "Importance of Events per Independent Variable in Proportional Hazards Regression Analysis II. Accuracy and Precision of Regression Estimates." *Journal of Clinical Epidemiology* 48 (12): 1503–10.

Vittinghoff, Eric, and Charles E Mcculloch. 2007. "Original Contribution Relaxing the Rule of Ten Events per Variable in Logistic and Cox Regression" 165 (6): 710–18.
<https://doi.org/10.1093/aje/kwk052>.

Appendix D

30 June 2020

Dear editors of *Royal Society Open Science*,

Please consider our revised submission entitled “**Evaluating how lethal management affects poaching of Mexican wolves**” (Manuscript ID RSOS-200330) as a registered report for *Royal Society Open Science*. We are grateful for the opportunity to revise and resubmit and have done our best to address the associate editor’s and reviewer’s latest comments and suggestions (**in bold below**).

Our proposed study will evaluate the effects of a widespread policy intervention on the hazard and incidence of illegal killing of a U.S. endangered species, the Mexican gray wolf (*Canis lupus baileyi*), in Arizona and New Mexico. We will use advanced biostatistical methods to analyze hazard, competing risks, and cumulative incidence functions for both disappearance (‘lost to follow-up’) and mortality endpoints. In short, we include disappearance as an endpoint alongside known-fate deaths and comparing endpoint rates in different, replicated policy periods. Including disappearance as an endpoint is novel in wildlife science and transformative because it takes advantage of recent insights that have falsified the assumption that marked animals that disappear can be ‘censored’ without biasing results.

This study would build upon and replicate methods developed on another population of endangered carnivores, the Western Great Lakes grey wolves. We test a hypothesis first published in The Proceedings of the Royal Society B in 2016 by Chapron & Treves [1]. That study found that during periods of down-listing or delisting under the U.S. Endangered Species Act, during which time states had greater authority to kill Wisconsin and Michigan wolves legally, there was a slow-down in both populations’ growths beyond the number of wolves legally killed. They inferred a new cause or dynamic of deaths of wolves, such as poaching, increased after ruling out reproduction or migration as probable causes. In a subsequent study in review, Santiago-Ávila 2019 found an increased hazard and incidence of disappearances of monitored Wisconsin gray wolves. Here we examine an analogous situation in the recovery zone of the Mexican gray wolf population.

We replicate a test of a hypothesis which triggered a great deal of controversy after the publication of the article in Proc. Roy. Soc. B cited above. We feel that having our methods evaluated as a registered report will help to identify concerns about methods and inferences early in the research process, while also reducing the likelihood of confirmation or rejection bias as advocated by the editors of Open Science in [2]. We welcome the added scrutiny and highest standards of scientific integrity implicit and explicit in the process for registered reports.

Our analyses address a growing interdisciplinary body of theory about the effects of liberalized killing on poaching risks and mortality patterns of endangered species [3-9]. The study period in our initial submission spanned 1998-2012, (n=168 radio-collared, adult Mexican wolves), with the possibility of extending it to 2016 if we obtained data on dead wolves currently under criminal investigation (potentially n=279). After our initial submission we continued with efforts to obtain such data spanning June 2012-December 2016, which the Office of Law Enforcement (OLE) withheld for legal reasons. During the interval between submission and revisions, we got confirmation of access to the withheld data. Therefore, we are pursuing our full data scenario (from 1998-2016) as we originally proposed. This expansion in our data should strengthen our inferences and contribute to further alleviate some of the reviewer’s concerns: now we have an

additional policy period in which liberalized killing began, so we have two replicated periods across hundreds of replicated largely independent wolves. We have also added Dave Parsons (Mexican Wolf Recovery Program Coordinator 1990-99) as a co-author, given his years of experience with the agency that collected these data and his intimate knowledge of their monitoring methods and policy shifts.

All necessary funding and approvals have already been acquired to complete the analysis.

We hereby certify we have not analyzed the data beyond what we present in the Registered Report and that the analysis will begin immediately upon provisional acceptance of the Stage 1 manuscript. Should this registered report be accepted and following stage 1, raw data, along with our protocols, will be made available on the Open Science Framework under private embargo until the submission of the Stage 2 manuscript.

As per the associated editor's and reviewer's latest constructive comments and suggestions, we have refined the proposal with special attention to:

- **Edited the explanation of BFs and specifications to address the reviewer's concerns and be explicit about the specifications used**
- **Incorporated robustness checks through different specifications of the predicted effect (rather than a robustness region) given scant evidence for any particular specification**
- **Clarifying how we will decide between supporting, contradictory and inconclusive evidence in terms of BFs**
- **Opted for including an analysis of the poa+ltf endpoint for comparison purposes and as a measure of an effect on 'total' poaching (following the reviewer's comment)**

We hope the latest revision fully addresses any remaining concerns. Thank you for considering and with kind regards,

Francisco J. Santiago-Ávila, Naomi Louchouart, David R. Parsons & Adrian Treves

References

1. G. Chapron, A. Treves, Blood does not buy goodwill: allowing culling increases poaching of a large carnivore. *Proc. R. Soc. London B Biol. Sci.* 283, 20152939 (2016).
2. J. Sanders, J. Blundy, A. Donaldson, S. Brown, R. Ivison, M. Padgett, K. Padian, K. Rittinger, K. Rowe, A. Stace, E. Viding, C. Chambers, M. Chaplain, Transparency and openness in science. *R Soc. Open Sci.* (2017) <https://doi.org/10.1098/rsos.160979>.
3. W. J. Ripple et al., Status and ecological effects of the world's largest carnivores. *Science* (80-.) 343, 1241484 (2014).
4. S. Creel et al., Questionable policy for large carnivore hunting. *Science* (80-.). 350, 1473–1475 (2015).

5. A. Treves, M. Krofel, J. V Lopez-Bao, Missing wolves, misguided policy. *Sci.* 350, 1473–1475 (2016).
6. G. Chapron, A. Treves, Reply to comments by Olson et al. 2017 and Stien 2017. *Proc. R. Soc. B Biol. Sci.* 284 (2017), doi:10.1098/rspb.2017.1743.
7. E. R. Olson et al., Pendulum swings in wolf management led to conflict, illegal kills, and a legislated wolf hunt. *Conserv. Lett.* 8, 351–360 (2015).
8. J. Suutarinen, I. Kojola, One way or another: predictors of wolf poaching in a legally harvested wolf population. *Anim. Conserv.*, 1–9 (2018).
9. J. Suutarinen, I. Kojola, Poaching regulates the legally hunted wolf population in Finland. *Biol. Conserv.* 215, 11–18 (2017).

Appendix E

Response to Reviewers and Editors

This is a Registered Report Stage 2 submission and has yet to be reviewed by reviewers and editors.

Appendix F

Response to Reviewers and Editors

We would like to thank the editor and reviewer for their comments and suggestions, and the opportunity to improve the manuscript. Below, we respond to reviewer #2, who has been very helpful and constructive, in regards to our disagreement with his suggestion of an ‘unplanned’ analysis section. We address each concern in detail to show why we demur.

Comments to the Author(s)

The manuscript is clear and well written. However, the authors need to strictly adhere to the logic of a registered report. Specifically the analyses need a section where the planned analyses are given, whether they pan out as appropriate or not; and then a different clearly labelled section "non-planned" or "exploratory", even if the non-planned analyses were well justified. "Non-planned" does not mean that the analyses are suspect; just that the reader needs to know they were not planned, including readers who just read e.g. abstracts or discussions.

Planned analyses referred to inference by Bayes Factors, specifically deciding according to whether 2 out of 3 pointed in a certain direction. The planned analysis section should stick to this without using p-values for inference. The DV should be the sum of POA+LTF. Then an exploratory analysis section can be introduced, that digs into different DVs more, as the authors have done, changes the use of BFs or uses p-values.

In the discussion first consider the planned analyses then make explicit what follows from non-planned analyses.

We admit we were puzzled by this interpretation and suggestion. We conceive of ‘planned analysis’ as referring to the entire Methods section as well as Table 4 (3rd column, ‘Analysis Plan’), rather than our decision criteria for hypotheses (Table 4, column 4, ‘Interpretation...’). These analyses include Cox and FG models for each endpoint, CIFs and finally BFs that inform our strength of evidence. We believe our Results section to be following strict reporting of these analyses required by registered reports and the competing risk literature, all of which were planned, described and referenced in Methods. We cannot identify any ‘unplanned’ analyses in our submission. In our results and discussion, we still report and discuss our models for all endpoints, including total potential poached, LTF and poached endpoints, and their BFs. Moreover, our methods did not state we would focus on POA+LTF, but that we would look at all three endpoints. Indeed, LTF+POA was explicitly added “for purposes of comparison” (p. 18, line 367) with the LTF and poached endpoints, rather than as the study focus.

We also clarify that we did not use p values for inference. Rather, we report them for those who wish to see them but base all of our inferences on BF as planned in the Methods.

To be more explicit in the text about the above, we have further clarified (in the ‘Results’ as well as ‘Discussion’) inference with BFs and any differences with our initial inference criterion in Table 4.

We understand how our interpretation of our analyses in our Discussion goes beyond that initial BFs criterion given issues with BF specifications, as already stated in our Discussion section.

Rather than having ‘planned’ and ‘unplanned’ analysis sections, which could confuse readers (as it did us), we hope the above explanation and proposed brief in-text clarifications can resolve the matter.

Do not assert no effect (lines 575, 613) without evidence for it (e.g. a relevant Bayes factor, even if unplanned)

Thank you for this observation. We have modified such statements in the abstract and main text to indicate we expected a greater change than observed in those parameters.

The abstract should first report conclusions from planned analyses and then indicate any other conclusions it reports follow from "exploratory" or "unplanned" analyses.

We have modified the abstract accordingly to indicate first the inconclusive evidence for reported poaching, and then the conclusive evidence for an increase in disappearances.

P 25: "Covariates of winter and sex did not significantly affect the results of any models, and therefore the most parsimonious model included the policy intervention without either covariate."

I could not find a statement in the planned analyses that non-significant co-variates would be removed. Thus, for the planned analyses I see no justification for removing these covariates. (They could be reported in a non-planned analyses section).

We did include a statement that we would avoid "including covariates unless they are essential to control." (p. 12) in our initial registered report, which is why we report univariate models in the main text (because excluded covariates were uninformative), but report multivariate results of uninformative covariates in our Suppl. Mat. (Table S1). However, given concerns over misinterpretation of the statement, we have attempted to clarify, as follows: "We have therefore excluded from our models any covariates unless they are essential to control." (p. 12, line 305)